# Neutralizing antibodies induced in immunized macaques recognize the CD4-binding site on an occluded-open HIV-1 envelope trimer

Zhi Yang[1], Kim-Marie A. Dam [1], Michael D. Bridges[2,3], Magnus A. G. Hoffmann [1], Andrew T. DeLaitsch [1], Harry B. Gristick[1], Amelia Escolano[4], Rajeev Gautam[5], Malcolm A. Martin[5], Michel C. Nussenzweig[4,6], Wayne L. Hubbell[2,3] & Pamela J. Bjorkman [1✉]

Broadly-neutralizing antibodies (bNAbs) against HIV-1 Env can protect from infection. We characterize Ab1303 and Ab1573, heterologously-neutralizing CD4-binding site (CD4bs) antibodies, isolated from sequentially-immunized macaques. Ab1303/Ab1573 binding is observed only when Env trimers are not constrained in the closed, prefusion conformation. Fab-Env cryo-EM structures show that both antibodies recognize the CD4bs on Env trimer with an 'occluded-open' conformation between closed, as targeted by bNAbs, and fully-open, as recognized by CD4. The occluded-open Env trimer conformation includes outwardly-rotated gp120 subunits, but unlike CD4-bound Envs, does not exhibit V1V2 displacement, 4-stranded gp120 bridging sheet, or co-receptor binding site exposure. Inter-protomer distances within trimers measured by double electron-electron resonance spectroscopy suggest an equilibrium between occluded-open and closed Env conformations, consistent with Ab1303/Ab1573 binding stabilizing an existing conformation. Studies of Ab1303/Ab1573 demonstrate that CD4bs neutralizing antibodies that bind open Env trimers can be raised by immunization, thereby informing immunogen design and antibody therapeutic efforts.

[1] Division of Biology and Biological Engineering, California Institute of Technology, Pasadena, CA, USA. [2] Jules Stein Eye Institute, University of California, Los Angeles, CA, USA. [3] Department of Chemistry and Biochemistry, University of California, Los Angeles, CA, USA. [4] Laboratory of Molecular Immunology, The Rockefeller University, New York, NY, USA. [5] Laboratory of Molecular Microbiology, National Institute of Allergy and Infectious Diseases, National Institutes of Health, Bethesda, MD, USA. [6] Howard Hughes Medical Institute, The Rockefeller University, New York, NY, USA. ✉email: bjorkman@caltech.edu

Human immunodeficiency virus-1 (HIV-1) is the causative agent behind the ongoing AIDS pandemic affecting millions of people worldwide. Although HIV-1 infection induces neutralizing antibodies against the viral envelope glycoprotein trimer (Env), the large number of viral strains in a single infected person and across the infected population means that commonly-produced strain-specific antibodies do not clear the infection[1]. However, a fraction of infected patients produce broadly neutralizing antibodies (bNAbs) that could provide protection from HIV-1 infection if an efficient means of eliciting such antibodies is developed[2,3]. However, vaccines to elicit such bNAbs are challenging to develop because heavily somatically mutated bNAbs usually arise only after years of virus-antibody co-evolution in their hosts[2,3].

Neutralizing antibodies against HIV-1 are exclusively directed against Env, the only viral protein on the surface of the virion[4,5]. HIV-1 Env is a homotrimer of gp120-gp41 heterodimers that mediates fusion of the host and viral membrane bilayers to allow entry of viral RNA into the host cell cytoplasm[6]. Fusion is initiated when the Env gp120 subunit contacts the host receptor CD4, resulting in conformational changes that reveal the binding site for a host coreceptor in the chemokine receptor family[7,8]. Coreceptor binding to gp120 results in further conformational changes including insertion of the gp41 fusion peptide into the host cell membrane[6].

Conformations of trimeric HIV-1 Envs have been investigated using single-particle cryo-EM to derive structures of soluble, native-like Env trimers lacking membrane and cytoplasmic domains and including stabilizing mutations (SOSIP.664 Envs)[9]. Such structures defined a closed, pre-fusion Env state in which the coreceptor binding site on gp120 variable loop 3 (V3) is shielded by the g120 V1V2 loops[10], and open CD4-bound Env trimer states with outwardly-rotated gp120 subunits and V1V2 loops displaced by ~40 Å to expose the V3 loops and coreceptor binding site[11–14].

Structurally-characterized anti-HIV-1 bNAbs recognize the closed, pre-fusion Env state[10] with the exception of one of the first HIV-1 bNAbs to be discovered: an antibody called b12 that was isolated from a phage display screen[15]. Like more recently identified bNAbs[16,17], b12 binds to an epitope overlapping with the CD4-binding site (CD4bs) on gp120[18]. However, the Env trimer state recognized by b12 represents an 'occluded-open' conformation in which the gp120 subunits are rotated out from the central trimer axis, but V1V2 is not displaced to the sides of the Env trimer[11,12,14].

As compared with a library screen that would not preserve correct heavy chain-light chain pairing, Ab1303 and Ab1573 were isolated by single cell cloning from SOSIP-binding B cells derived from sequentially-immunized non-human primates (NHPs)[19]. Both antibodies exhibited broad, but weak, heterologous neutralization and were mapped by competition ELISA as recognizing the CD4bs[19]. Here we show that, in common with b12 but not with other CD4bs bNAbs, neither antibody binds Env trimer when it is locked into the closed, prefusion state that is recognized by other CD4bs bNAbs. To elucidate the conformational state of Env recognized by these monoclonal antibodies (mAbs), we solved single-particle cryo-EM structures of a SOSIP Env trimer complexed with either Ab1303 or Ab1573 Fabs. The structures revealed that these mAbs recognized Env trimers with gp120 subunits that had rotated outwards to create an occluded-open trimer conformation that differed from the closed, prefusion Env conformation and from the open conformation of CD4-bound Env trimers[11–14]. To further investigate the occluded-open Env trimer conformation, we used double electron-electron resonance (DEER) spectroscopy to determine if this conformation was detectable in a solution of unliganded HIV-1 Env trimers. By measuring inter-protomer distances between V1V2 loops in the presence and absence of Ab1303 and Ab1573, we found evidence for the conformation recognized by these antibodies in unliganded trimers, suggesting that Ab1303 or Ab1573 binding stabilized a pre-existing Env conformation.

In this work, in contrast to previous structures of CD4bs bNAb-closed Env trimer complexes[10] and CD4-bound open Env conformation structures[11–14], the Ab1303 and Ab1573 structures revealed a new mode of naturally-induced CD4bs antibody-Env interaction. Furthermore, when combined with DEER spectroscopy data, these structures define Env trimer conformational state intermediates between the closed and CD4-bound open conformations. Although Ab1303 and Ab 1573 are not as broad or potent as CD4bs bNAbs isolated from infected human donors, discovery of the Env structure recognized by these mAbs reveals an unanticipated target that could be exploited for immunogen design.

## Results

### Heterologously neutralizing mAbs Ab1303 and Ab1573 were elicited in NHPs after sequential immunization with designed immunogens.

The V3-glycan patch immunogen RC1 was modified from the V3 immunogen 11MUTB[20] by mutating a potential N-linked glycan site (PNGS) to remove the N-glycan attached to $Asn156_{gp120}$[21]. RC1 and 11MUTB were both derived from the clade A BG505 SOSIP.664 native-like Env trimer[9]. We constructed RC1-4fill and 11MUTB-4fill by modifying RC1 and 11MUTB, respectively, to insert PNGSs to add glycans to residues $230_{gp120}$, $241_{gp120}$, $289_{gp120}$, and $344_{gp120}$ to reduce antibody responses to off-target epitopes[22–24]. Immunogens were multimerized on VLPs using the SpyTag-SpyCatcher system[25,26] to enhance avidity effects and limit antibody access to the Env trimer base. The mAbs described here were isolated from NHPs immunized sequentially as shown in Supplementary Fig. 1a. As described elsewhere[19], we obtained weak heterologously neutralizing antisera from the sequentially-immunized NHPs, and mAb sequences were generated by single cell cloning from B cells that were captured as described using BG505 and B41 SOSIP baits[27]. Here we investigated Ab1303 and Ab1573, which unexpectedly recognized the CD4bs rather than the V3-glycan patch that was targeted in the sequential immunization scheme.

Ab1303 sequences were derived from rhesus macaque germline V gene segments IGHV4-160*01 and IGLV4-97*01 gene segments for heavy and light chains, respectively, and exhibited 8.3% and 8% amino acid changes due to somatic hypermutations, respectively (Supplementary Fig. 1b). Ab1573 sequences were derived from IGHV1-198*02 and IGLV1-64*01 gene segments and contained 7.3% and 10.5% somatic hypermutation changes, respectively (Supplementary Fig. 1b). Neutralizing activities of the two antibodies were reported elsewhere[19]. The HIV-1 strains chosen for neutralization measurements were derived from the 12-strain global panel of HIV-1 reference strains[28] plus seven other HIV-1 strains including BG505, from which the RC1-4fill and 11MUT-4fill immunogens were derived. We found that Ab1303 neutralized 12 of the 19 cross-clade strain panel with $IC_{50}$ values < 100 μg/mL, whereas Ab1573 neutralized five strains in the panel with $IC_{50}$ values < 100 μg/mL[19]. Although the neutralization potencies were generally weak, both mAbs exhibited heterologous neutralization, with Ab1303 neutralizing >60% of the viruses in the cross-clade panel when evaluated at high concentrations.

### Ab1303/Ab1573 bound a non-closed Env trimer conformation with varying stoichiometries.

To verify that Ab1303 and Ab1573 recognized the CD4bs on Env trimer, we repeated competition

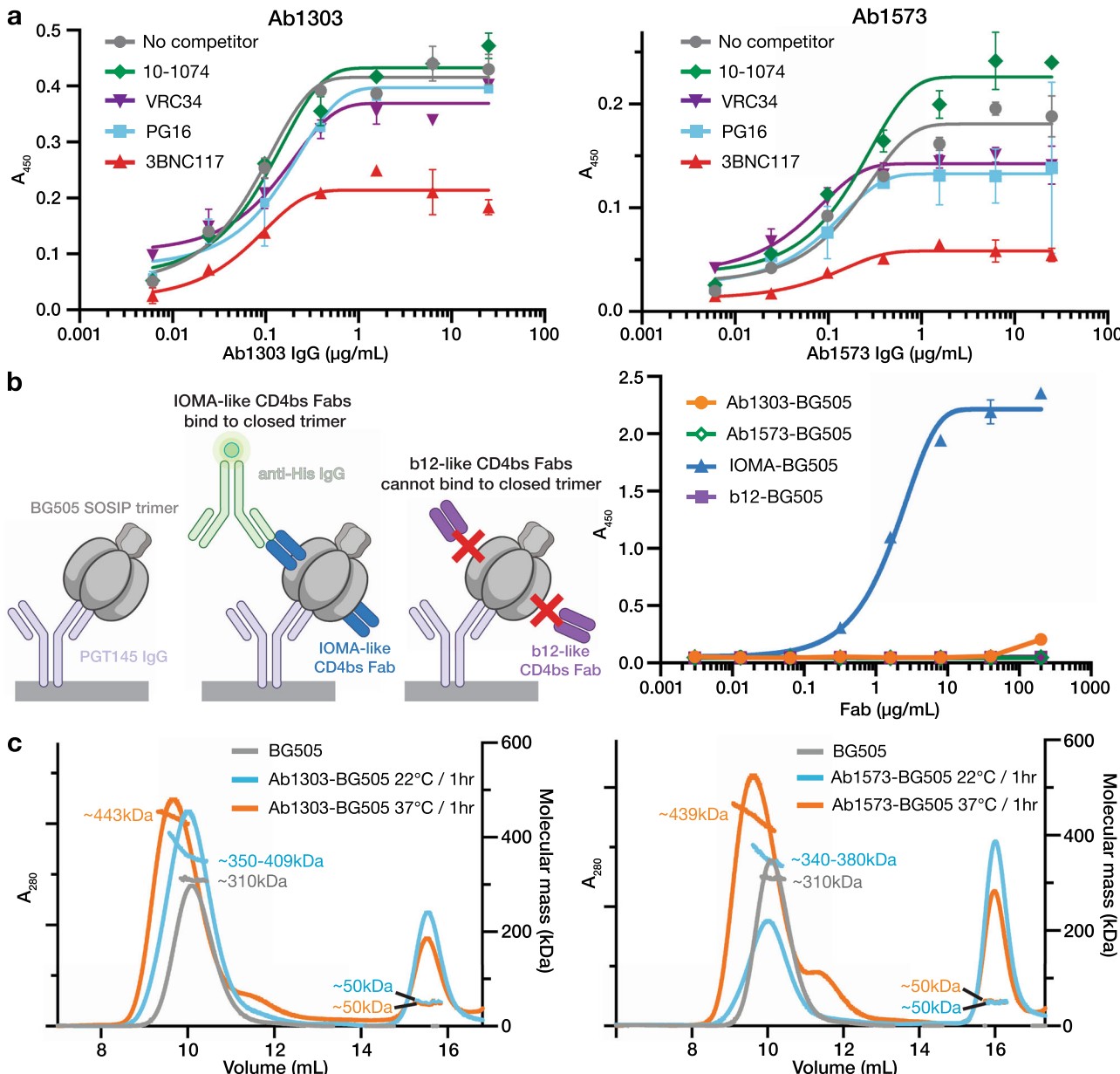

**Fig. 1 Ab1303 and Ab1573 are CD4bs NAbs that bind an Env conformation other than the closed, prefusion state. a** Competition ELISA to map the binding sites of Ab1303 (left) and Ab1573 (right) on BG505 SOSIP. Randomly biotinylated BG505 SOSIP trimer was immobilized on a streptavidin plate and then incubated with a Fab from a bNAb that recognizes the V3 loop (10-1074, green, diamonds), the fusion peptide (VRC34, purple, inverted triangles), the V1V2 loop (PG16, blue, squares), or the CD4bs (3BNC117, red, triangles). Increasing concentrations of Ab1303 or Ab1573 IgG were added to the Fab-BG505 complex, resulting in competition in both cases with the CD4bs bNAb 3BNC117. Data represents mean ± s.d. from $n = 2$ biologically independent samples. Source data are provided as a Source Data file. **b** ELISA to assess binding of Ab1303 and Ab1573 to closed, prefusion conformation of BG505 trimer. Left: schematic of the experiment: PGT145 IgG (light purple), which constrains BG505 to a closed prefusion conformation, was immobilized and captured BG505 (gray). Binding of His-tagged Fab (blue or dark purple) to BG505 was detected using a labeled anti-His-tag antibody (light green). Representation is created with BioRender.com. Right: Ab1303 (orange), Ab1573 (green), and b12 (dark purple) do not bind to BG505 captured by PGT145, but IOMA (blue) binds to BG505 captured by PGT145. Data represents mean ± s.d. from $n = 2$ biologically independent samples. Source data are provided as a Source Data file. **c** SEC-MALS of Ab1303-BG505 (left) and Ab1573-BG505 (right) complexes incubated at different temperatures. SEC traces were monitored by absorption at 280 nm. Traces and estimated molecular mass distributions for unliganded BG505 (gray) and for Ab1303 Fab or Ab1573 Fab incubated with BG505 SOSIP trimer at 37 °C (orange) or 22 °C (blue) for 1 h are shown.

ELISA experiments conducted with RC1 trimer[19], this time using BG505 trimer (Fig. 1a). We first immobilized randomly-biotinylated BG505 Env trimers on streptavidin plates and then added antibody Fabs targeting either the CD4bs (3BNC117)[16], the V3-glycan patch (10-1074)[29], V1V2 (PG16)[30], or the fusion peptide (VRC34)[31]. Subsequently, either Ab1303 or Ab1573 IgGs were added, the plates were washed, and the degree of binding

was detected. The binding of Ab1303 IgG was essentially unaffected in the Env trimer samples that were complexed with 10-1074, PG16, or VRC34 compared to the control with no competitor (Ab1303 IgG alone), but its binding to BG505 Env was reduced in the presence of 3BNC117 Fab (Fig. 1a). Similar results were found for Ab1573, although the presence of PG16 and VRC34 Fabs also reduced the binding somewhat (Fig. 1a). From

these results, we concluded that both antibodies recognized the CD4bs on BG505 SOSIP.

CD4bs bNAbs such as VRC01, 3BNC117, and IOMA bind closed, prefusion state Env trimers[32–34]. An exception to this finding for CD4bs antibodies is b12, a more weakly neutralizing antibody selected from a phage display derived from antibody genes isolated from an HIV-positive individual bone marrow[15]. Unlike all other human CD4bs bNAbs characterized to date, b12 binds to an "occluded open" Env trimer conformation in which gp120 subunits are rotated outwards from the central trimer axis but the V1V2 loops are not displaced from their positions on top of the gp120 subunits[11].

To determine if Ab1303 and Ab1573 recognize closed Env trimers, we assessed their ability to bind Env trimer captured with PGT145, a V1V2 bNAb that recognizes a quaternary epitope at the trimer apex[33]. Unlike other V1V2 bNAbs such as PG16 that can recognize both closed and open Env trimers[35], PGT145 locks Envs into a closed, prefusion state[33]. In this experiment, we captured BG505 with PGT145 IgG on an ELISA plate and compared binding of Ab1303, Ab1573, a conventional CD4bs bNAb that recognizes closed Env trimer (IOMA)[32], and b12 Fab that binds to an "occluded open" trimer (Fig. 1b). IOMA showed binding to BG505 captured by PGT145, consistent with IOMA-BG505 complex structure with a closed-conformation trimer[32] (Fig. 1b). By contrast, b12 did not bind to BG505 that was captured by PGT145, as the b12 epitope is occluded in the closed Env conformation[11] (Fig. 1b). Similar to the results for b12, Ab1303 and Ab1573 did not bind BG505 that was captured by PGT145, suggesting that the closed Env trimer occludes epitopes for Ab1303 and Ab1573 (Fig. 1b).

To further characterize the interactions of Ab1303 and Ab1573 with Env trimer, we performed size-exclusion chromatography (SEC) coupled with multi-angle light scattering (SEC-MALS) to determine the absolute molecular masses of the complexes and therefore the number of Fabs bound per trimer. Complexes formed by incubating mAb Fabs with BG505 trimer at various temperatures were analyzed by SEC-MALS. Compared to BG505 trimer alone (~310 kDa apparent mass including glycans), incubation of Ab1303 Fab with trimer at 22 °C for 1 h resulted in a complex with an average molecular mass of 370 kDa, equivalent to ~1.1 Fabs per trimer, whereas incubation at 37 °C for 1 h produced complexes with an average molecular mass of 443 kDa, corresponding to ~3 Fabs per trimer (Fig. 1c). In the case of Ab1573 complexes, 1 h incubations at 22 °C and 37 °C produced up to ~1 and ~2.6 copies of Fabs per trimer on average (Fig. 1c). Notably, peaks corresponding to the Ab1303 Fab-BG505 complex from the 22 °C incubation condition and the Ab1573 Fab-BG505 complexes from both temperature conditions were broad, consistent with a mixture of sub-stoichiometric populations being present under these conditions. These observations suggest that physiological temperature could result in the antibody binding sites on Env being more accessible, facilitating binding of Ab1303 and Ab1573 by more frequent Env transitions between different conformational states at higher temperature.

**Ab1303 and Ab1573 occlude the CD4bs on gp120.** To further explore the interactions of Ab1303 and Ab1573 with Env trimer, we solved 1.51 Å and 2.50 Å crystal structures of unbound Ab1303 and Ab1573 Fabs (Supplementary Table 1; Supplementary Fig. 1c) and single-particle cryo-EM structures of BG505 SOSIP.664 complexed with Ab1303 and Ab1573 to resolutions of 4.0 Å and 4.1 Å, respectively (Fig. 2a, b, Supplementary Fig. 2, and Supplementary Table 2). Prior to cryo-EM data collection, the Fabs were incubated with BG505 at 37 °C for two hours to achieve an ~3:1 Fab to BG505 trimer stoichiometry.

We first compared the structures of the unbound Fabs to their structures when bound to Env trimer. For both antibodies, there were no major structural changes between their free (solved by X-ray crystallography) and bound (solved by cryo-EM) forms: Root mean square deviations, rmsds, for superimposition of free and bound Ab1303 $V_H$ and $V_L$ (235 Cα atoms) were 0.72 Å, and 0.89 Å for superimposition of free and bound Ab1573 $V_H$ and $V_L$ (231 Cα atoms), with minimal differences in the complementarity determining region (CDR) loops (Supplementary Fig. 1d). Thus, Ab1303 and Ab1573 bound their Env antigen targets using preformed antibody combining sites, rather than undergoing structural rearrangements to accommodate their targets.

The cryo-EM structures of both Fab complexes with Env trimer showed three bound Fabs that interacted with the CD4bs of each gp120 protomer (Fig. 2a, b; Fig. 3). The binding sites for Ab1303 and Ab1573 on gp120 were located in an area that is surrounded by three N-glycan patches: the $Asn363_{gp120}$/$Asn386_{gp120}$ glycans located near the base of V3, the $Asn197_{gp120}$ glycan in the V1V2 region, and the $Asn276_{gp120}$ glycan near the bottom of gp120 (Fig. 2a, b). The epitope of Ab1303 comprised 1430Å² of buried surface area (BSA) on gp120, of which 879Å² were buried by the heavy chain and 551Å² were buried by the light chain (Fig. 3a, g). The heavy chain CDR 3 (CDRH3) of Ab1303 makes extensive contacts with gp120, including an antibody residue $Arg100B_{HC}$ that is stabilized by neighboring $Trp100A_{HC}$ through cation-π interaction, contributes a salt bridge with gp120 residue $Asp457_{gp120}$ and hydrogen bonds with the $Arg456_{gp120}$ carbonyl group and with the $Ser365_{gp120}$ side chain, forming a stable interaction network (Fig. 3b). Residue $Tyr100E_{HC}$ of CDRH3 hydrogen bonded with $Gln428_{gp120}$ and $Asp474_{gp120}$ (Fig. 3b). Residue $D368_{gp120}$ in the CD4 binding loop, which plays an important role in CD4 binding[36], was positioned in proximity to the Ab1303 light chain and excluded from solvent by adjacent CDRL3 residues such as $Trp91_{LC}$. In the case of Ab1573, 646Å² of surface area was buried by $V_H$ and 762Å² buried by $V_L$, composing a total of 1408Å² of BSA on gp120 (Fig. 3c, g). Two salt bridges were found at the Ab1573-gp120 interface: between CDRH3 residue $Asp97_{HC}$ and $Arg476_{gp120}$, and between CDRH1 residue $Arg31_{HC}$ and $Asp113_{gp120}$ (Fig. 3d). Compared with the Ab1573 and Ab1573 footprints on gp120, contacts by b12 are dominated by its $V_H$ domain (Fig. 3e), with no contacts by $V_L$ except for a small region of BSA (340 Å²) on V1V2 (Fig. 3f, g). A large portion of CD4 binding loop including residue $Asp368_{gp120}$ was positioned between Ab1573 CDRH3 and CDRL3.

In addition to their protein epitopes, Ab1303 and Ab1573 also contacted N-linked glycans on gp120. The light chains of both Fabs were adjacent to the N-glycans attached to $Asn197_{gp120}$ and $Asn276_{gp120}$. When compared to an Env structure that does not have an antibody bound in this vicinity (PDB 5FYL), the $Asn197_{gp120}$ side chain was rotated ~180° in the Ab1303- and Ab1573-bound structures, and the $Asn197_{gp120}$ glycans underwent large displacements. In addition, the $Asn276_{gp120}$ glycan was shifted slightly to facilitate Fab binding (Supplementary Fig. 3).

To further characterize the antibody epitope, we compared our structures with those of other CD4bs-Env complexes. The $V_H$ domains of Ab1303 and Ab1573 were positioned close to the gp120 inner domain, which is not fully exposed in the closed Env conformation, whereas the $V_L$ domains were sandwiched between $Asn363_{gp120}$/$Asn386_{gp120}$ and $Asn276_{gp120}$ glycans (Fig. 4). While the b12 $V_H$ is positioned similarly on gp120 as the $V_H$ domains of Ab1303 and Ab1573 (Fig. 3a, c, e), the b12 $V_L$ is closer to the gp120 V1V2, which constitutes the only contacts made by the b12 $V_L$ with gp120 (Fig. 3e–g). By contrast, other CD4bs bNAb $V_H V_L$ domains share similar binding poses; they are located further from the gp120 inner domain and are lined up almost parallel to

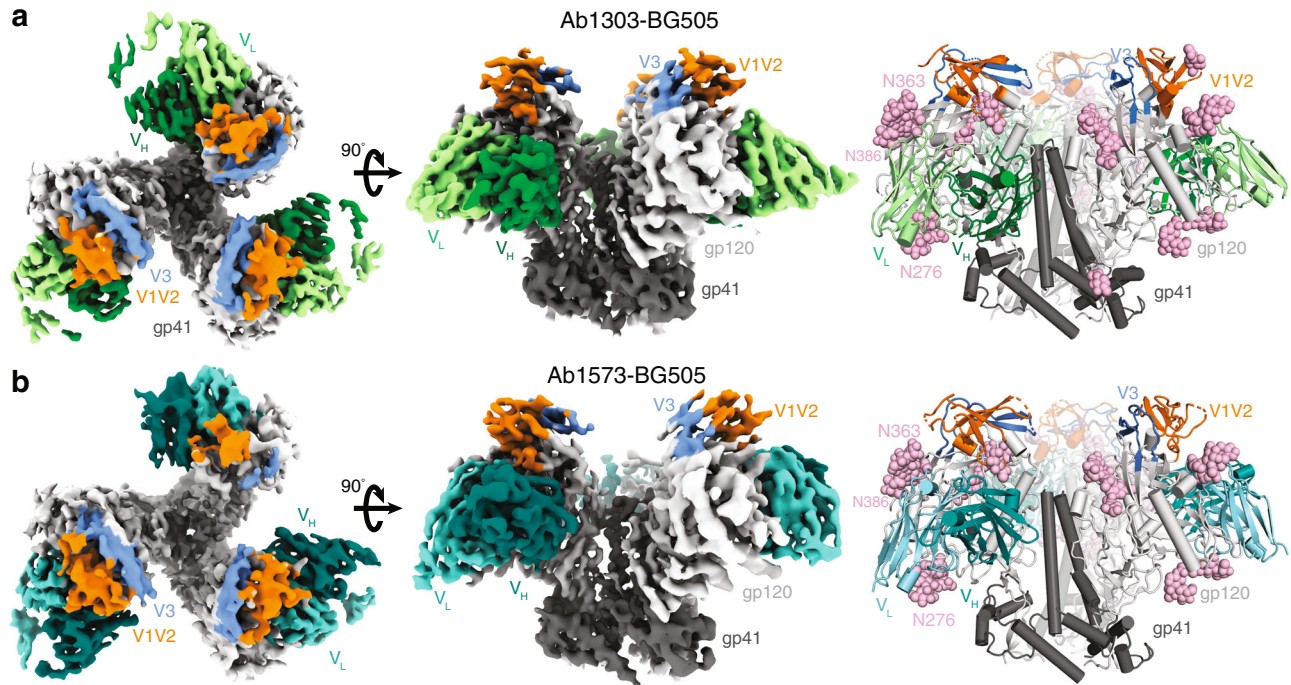

**Fig. 2 Cryo-EM maps and atomic models of Ab1303-BG505 and Ab1573-BG505 complexes. a, b** Cryo-EM density maps of Ab1303-BG505 (**a**) and Ab1573-BG505 (**b**) complexes shown from the top (left) and side (middle). Right: cartoon diagrams of the structures with *N*-linked glycans shown as pink spheres. Highlighted in colors include: gp120 subunits (light gray), gp41 (dark gray), Ab1303 VH (dark green) and VL (pale green) domains and Ab1573 VH (dark teal) and VL (light teal) domains. The gp120 V1V2 and V3 regions are shown in orange and blue, respectively.

the trimer threefold axis, with $V_H$ near the $Asn363_{gp120}$/$Asn386_{gp120}$ and $Asn197_{gp120}$ glycans and $V_L$ adjacent to the $Asn276_{gp120}$ glycan (Fig. 4), and to accommodate such poses, Asn276 glycans need to be displaced further away by antibodies light chains.

**Ab1303 and Ab1503 bind an occluded-open state of Env trimer**. The trimers in the Ab1303-Env and Ab1573-Env complexes differed in conformation from the closed, prefusion Env, each exhibiting a more open state that exposed portions of the gp120 that were otherwise buried (Fig. 5a, b). To characterize these differences, we mapped the trimer epitope regions from each antibody-bound open conformation onto a closed, prefusion trimer Env structure. For both complexes, a portion of the epitope was solvent inaccessible in the closed trimer state but was exposed in the antibody-bound open state (Fig. 5a, b; red highlighted regions). The Ab1303 contacts that are buried in a closed trimer were contacted exclusively on the occluded, open trimer by its $V_H$ domain (Fig. 5a), which buried 286 Å$^2$ of gp120 surface area that would be inaccessible on a closed trimer. The contact residues buried in the closed Env state by Ab1573 also involved only its $V_H$ domain (Fig. 5b), burying a discontinuous 137 Å$^2$ of gp120 surface area. In addition, docking of Ab1303 (Fig. 5c) or Ab1573 (Fig. 5d) onto a closed trimer structure results in steric clashes. These results are consistent with the ELISA demonstration that Ab1303 and Ab1573 did not bind the closed, prefusion Env trimers (Fig. 1b).

To compare and quantify outward displacements of gp120 protomers in different Env states, we measured inter-protomer distances of selected residues located in the CD4bs, V1V2 base, and V3 base of a closed BG505 Env trimer with analogous residues in BG505 trimers bound to Ab1303 or Ab1573 (Fig. 5e). Inter-protomer distances were increased in the Ab1303- and Ab1573-bound trimers compared with the closed trimer, providing a quantitative measurement of openness. In addition,

differences in the three inter-protomer distances for each measurement within the Ab1303- and Ab1573-bound Envs demonstrated trimer asymmetry compared with the symmetric closed trimer conformation. Comparisons with a structure of B41 SOSIP bound to b12 Fab[11] showed that the Ab1303- and Ab1573-bound trimers resembled the b12-bound Env state more than the closed state, although the b12-bound trimer structure was threefold symmetric (Fig. 5e). Finally, measurements for all four of these trimers differed from the CD4-bound open state exemplified by the structure of a CD4-bound asymmetrically open BG505 trimer[14] and a CD4-bound symmetrically-open B41 trimer[11] (Fig. 5e).

Despite Env trimer opening, the gp120 V1V2 and V3 regions in the Ab1303-BG505 and Ab1573-BG505 structures exhibited only minor local structural rearrangements in which the gp120s were displaced as nearly rigid bodies from their central positions in the closed trimer structure (Fig. 6a; Movie S1). Thus, most of each gp120 subunit, including the V1V2 and V3 regions, remained unchanged between the closed Env conformation and the open Ab1303- and Ab1573-bound Env trimers (Fig. 6a, left and middle); the rmsd for superimposition of 347 gp120 Cα atoms ($Gly41_{gp120}$ to $Pro493_{gp120}$ excluding disordered residues) was ~1.3 Å. By contrast, when bound to CD4, Env trimer opening did not result from rigid body rotations of gp120; instead the V1V2 loops were displaced from apex of each gp120 apex to the sides of the Env trimer to expose the coreceptor-binding site on V3, and portions of V1V2 and V3 were disordered[11,12,14] (Fig. 6a, right). In addition, the gp120 β2 and β3 strands at the beginning and end of the V1V2 loop switched positions with respect to their locations in closed, prefusion Env trimers to form a four-stranded anti-parallel β-sheet (4-stranded bridging sheet) (Fig. 6b, right) instead of the 3-stranded sheet in closed Env structures (Fig. 6b; left) (Movie S1). Notably, the 'occluded open' Env trimer conformations observed upon binding of Ab1303, Ab1573, or b12 included the 3-stranded sheet found in the closed, prefusion

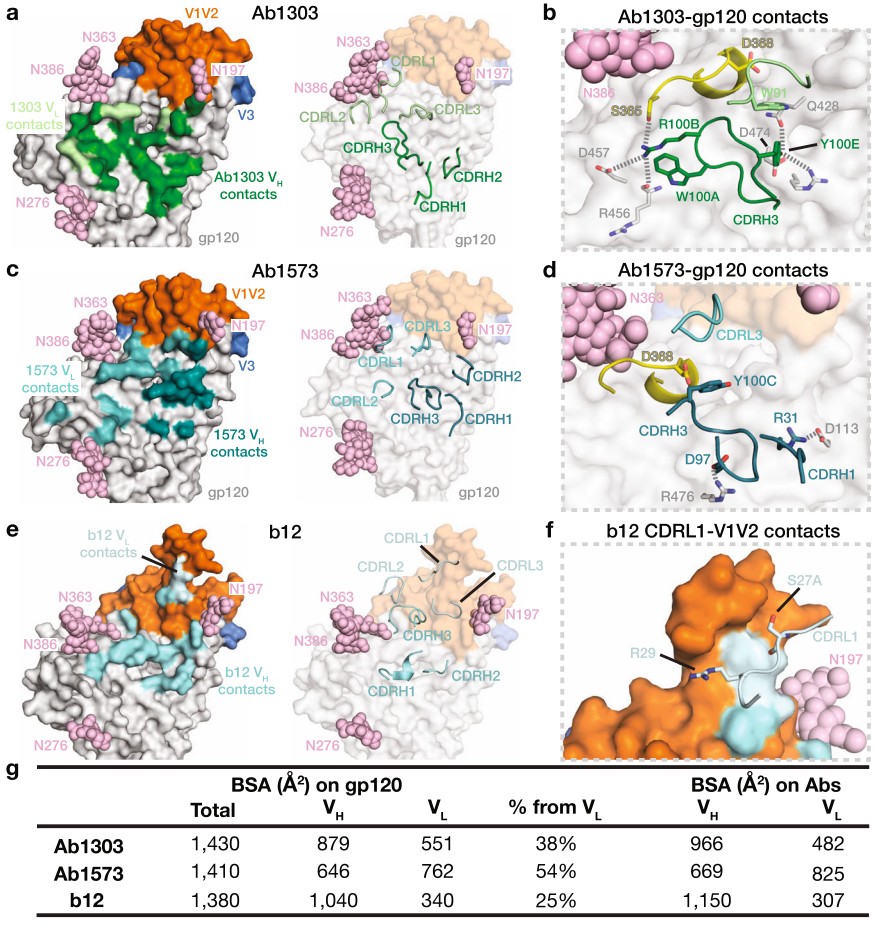

**Fig. 3 Ab1303 and Ab1573 recognize the CD4bs of BG505 Env trimer. a** Left: epitope of Ab1303 on gp120 surface with contacts made by antibody heavy and light chains colored in dark and light green, respectively. Right: CDR loops of Ab1303 mapped onto the gp120 surface. **b** Highlighted interactions between gp120 residues (gray sticks) and the Ab1303 CDRH3 (green sticks). The gp120 CD4 binding loop is shown in yellow with D368$_{gp120}$ residue highlighted in stick representation. **c** Left: epitope of Ab1573 on gp120 surface with contacts made by antibody heavy and light chains colored in dark and light teal, respectively. Right: CDR loops of Ab1573 mapped onto the gp120 surface. **d** Highlighted interactions between gp120 residues (gray sticks) and the Ab1573 CDRH1 and CDRH3 (teal sticks). CD4 binding loop of gp120 is shown in yellow with D368 residue shown in stick. **e** Left: epitope of b12 on gp120 surface with contacts made by antibody heavy and light chains colored in dark and light cyan, respectively. Right: CDR loops of b12 mapped onto the gp120 surface. **f** Highlighted interactions between gp120 V1V2 and the b12 V$_L$. **g** Comparison of the buried surface areas (BSAs) between the antibodies and gp120.

Env trimer rather than the 4-stranded bridging sheet in open, CD4-bound Env trimers (Fig. 6b; middle; Movie S1).

Local structural rearrangements in Asp57$_{gp120}$ - Ala73$_{gp120}$, residues that are immediately N-terminal to the gp120 α$_0$ region (residues Lys65$_{gp120}$ - Ala73$_{gp120}$), provided further evidence that the Ab1303- and Ab1573-bound trimers adopted a state distinct from the closed state recognized by CD4bs bNAbs: in the closed state, residues Asp57$_{gp120}$ - Glu62$_{gp120}$ formed a β-strand and a short loop (Fig. 6c, left), whereas they formed a two-turn α-helix and the α$_0$ residues remained as a loop in Ab1303/Ab1573-bound open conformation (Fig. 6c, middle). By contrast, in the CD4-bound fully-open state, the Asp57$_{gp120}$ - Glu62$_{gp120}$ segment formed a β-strand and short loop; whereas the α$_0$ segment (residues Lys65$_{gp120}$ - Ala73$_{gp120}$), which was a loop in both the closed and the Ab1303/Ab1573-bound open states, was an α-helix in the CD4-bound fully-open state (Fig. 6c, right). The structural rearrangement of α$_0$ was accompanied by protein side chain repositioning: in the closed and Ab1303/1573-bound open states, the Trp69$_{gp120}$ side chain was sandwiched between the α$_1$ helix and β$_4$ strand and His66$_{gp120}$ was solvent exposed, whereas in the CD4-bound open state, the Trp69$_{gp120}$ side chain was rearranged

such that the His66$_{gp120}$ side chain occupied a nearly analogous position (Fig. 6c).

The conformation of the C-terminal portion of gp41 heptad repeat segment 1 (HR1$_C$) also exhibited changes between the closed, Ab1303- and Ab1573-bound open, and CD4-bound open Env trimer states. In closed, prefusion Env trimers, residues N-terminal to Thr569$_{gp41}$ adopted a loop structure (Fig. 6c, left). Outward rotations of the gp120 subunits in the Ab1303-/Ab1573-bound open Env created space for the gp41 HR1$_C$ to extend its three-helix coiled-coil structure, lengthening the α-helices by 1.5 turns (Fig. 6c, middle). Additional outward gp120 rotations combined with V1V2 and V3 displacements created more space for the central α-helices in the CD4-bound fully-open state; thus gp41 residues that were disordered in the closed and occluded open trimer states extended the HR1$_C$ N-terminal helical structure by another helical turn, with ordered residues terminating at around residue Pro559$_{gp41}$[11–14], the site of the Ile-to-Pro stabilizing mutation in SOSIPs[9] (Fig. 6c, right). Thus, the Ab1303- and Ab1573-bound Env structures revealed an occluded-open trimer state distinct from both the closed, prefusion and the CD4-bound fully-open trimer conformations.

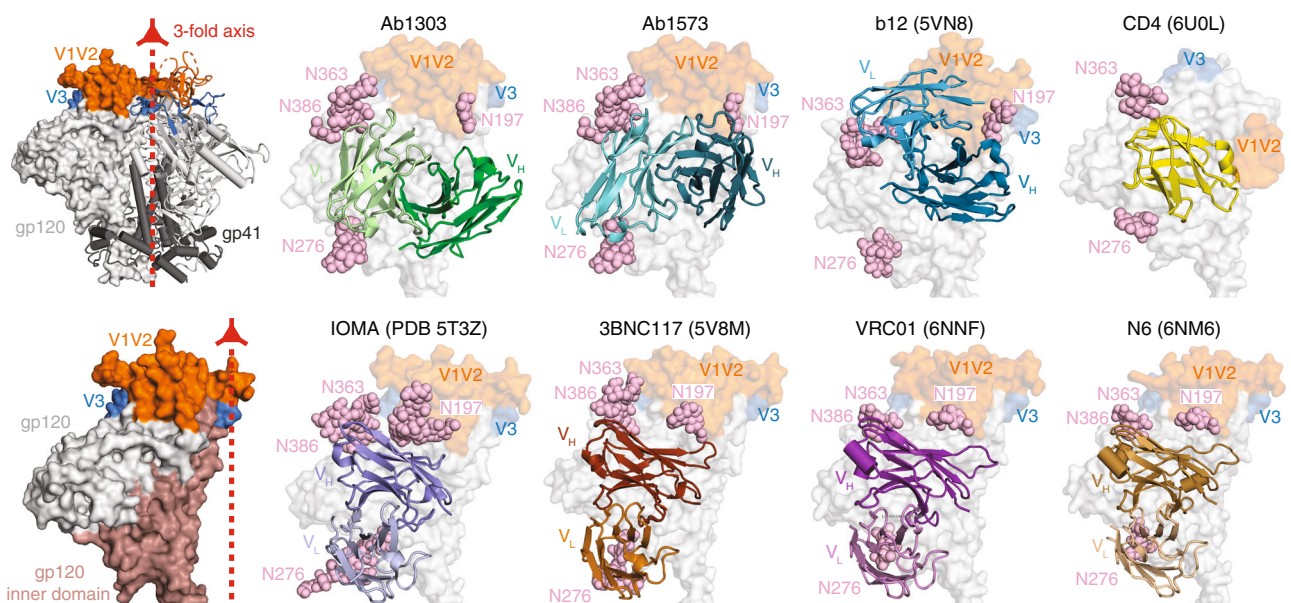

**Fig. 4 Differences in antibody-binding poses for CD4bs bNAbs compared with CD4.** Upper and lower left: surface representation of HIV-1 Env trimer (top) and gp120 monomer (bottom) showing locations of the threefold axis relating Env protomers (red arrow), the gp120 inner domain, V1V2, V3, and gp41. Remaining panels: Cartoon diagrams of $V_H$-VL domains of the indicated bNAbs bound to HIV-1 gp120 (gray surface, pink spheres for *N*-linked glycans, V1V2 in orange, V3 in blue). PDB codes: Ab1303 and Ab1573 (this study), b12 (5VN8), CD4 (6U0L), IOMA (5T3Z), 3BNC117 (5V8M), a VRC01 derivative (6NNF), and an N6 derivative (6NM6). The V1V2 and V3 regions were largely disordered in the CD4-bound Env structure (PDB 6U0L).

**DEER suggests that the unliganded Env trimers contain both occluded-open and other Env conformations.** To evaluate the conformational flexibility of ligand-free and antibody-bound trimer in solution, we used DEER spectroscopy to probe inter-protomer distances between V1V2 regions in different Env trimer states. DEER can be used to derive distances between electron spin pairs ranging from 17 to 80 Å by detecting their respective dipolar interactions[37]. By recording a snapshot of the equilibrium distance distributions of flash-frozen samples, DEER data report molecular motions in solution to provide insight into conformational heterogeneity. We previously used DEER to evaluate spin-labeled BG505 and B41 SOSIPs in the presence and absence of antibodies, CD4, and a small molecule ligand, finding a relatively homogeneous trimer apex, more conformational heterogeneity at the trimer base, and inter-protomer distances between spin labels that were consistent with bNAb-bound closed Env structures and CD4-bound open Env structures[38].

In the present studies, we introduced a free cysteine into a gp120-gp41 protomer of the BG505 SOSIP in order to use site-directed spin labeling[39] to covalently attach a nitroxide spin label with a V1 side chain[40]. This approach results in three spin labels on each Env, which form a triangle of spin labels, either equilateral, isosceles, or scalene depending on whether the labeled Env adopts a threefold symmetric or asymmetric conformation. Thus, DEER measurements in a conformationally-rigid Env trimer would report one distance in a symmetric Env and two or more distances in asymmetric Envs. The most probable distance in a DEER distribution is defined by the largest peak area and represents the dominant structural state in a population of states. The presence of multiple peaks in a DEER distribution indicates conformational heterogeneity, with individual peak widths related to the flexibility of that conformation and of the attached spin label[39,41]. In general, peaks representing 17–65 Å distances can be assigned with confidence, whereas distances >65 Å are detected with less accuracy[37].

To choose a site for spin labeling, we used the Ab1303- and Ab1573-Env structures to identify solvent-exposed residues, which when spin-labeled, would result in distinguishable inter-protomer distances in different Env conformations. We also restricted candidate sites to residues located in a β-sheet to minimize potential flexibility of the attached spin label and excluded residues that were involved in interactions with other residues to avoid disrupting protein folding. The optimal candidate residue, V1V2 residue Ser174$_{gp120}$, fulfilled these criteria, with inter-protomer Cα-Cα distances measured as 38 Å in a closed Env structure, ranging from 40 Å–60 Å in the asymmetric Ab1303- and Ab1573-bound Env structures, 67 Å in a b12-bound Env, and ~157 Å (far out of DEER range) in a CD4-bound open Env (Fig. 7a). Although the V1 spin label is small (about the size of an amino acid) and contributes limited width to DEER distance distributions[42], distances between spin label side chains measured by DEER only rarely equal the Cα-Cα inter-protomer distance since the radical center is found on the nitroxide ring, not the peptide linkage. As such, DEER results can be complicated by conformational heterogeneity and flexibility intrinsic to the protein studied. In addition, previous work to model V1 nitroxide side chain rotamers on BG505 Env DEER target sites suggested that differences in V1 rotamers can contribute to measured DEER distances[38].

The S174C mutant version of BG505 SOSIP was expressed, purified, and labeled with the V1 side chain. Ab1303 and Ab1573 Fabs were added at a three-molar excess to V1-labeled BG505, and liganded and unliganded samples were incubated at 37 °C for 3 h before being flash-frozen in liquid nitrogen for resonance measurements. The DEER spectrum of unliganded BG505 (black trace in Fig. 7b, c) showed a complicated collection of peaks, indicating conformational heterogeneity of the V1V2 region in the vicinity of gp120 residue 174. These results differ from previous DEER experiments from which we derived BG505 V1V2 distances from spectra recorded after incubation at 4 °C, which showed a more homogeneous distance distribution with a dominant peak observed at the expected inter-protomer distance[38]. In the present experiments, one of the major peaks, centered at ~38 Å, corresponds to the residue 174 inter-protomer

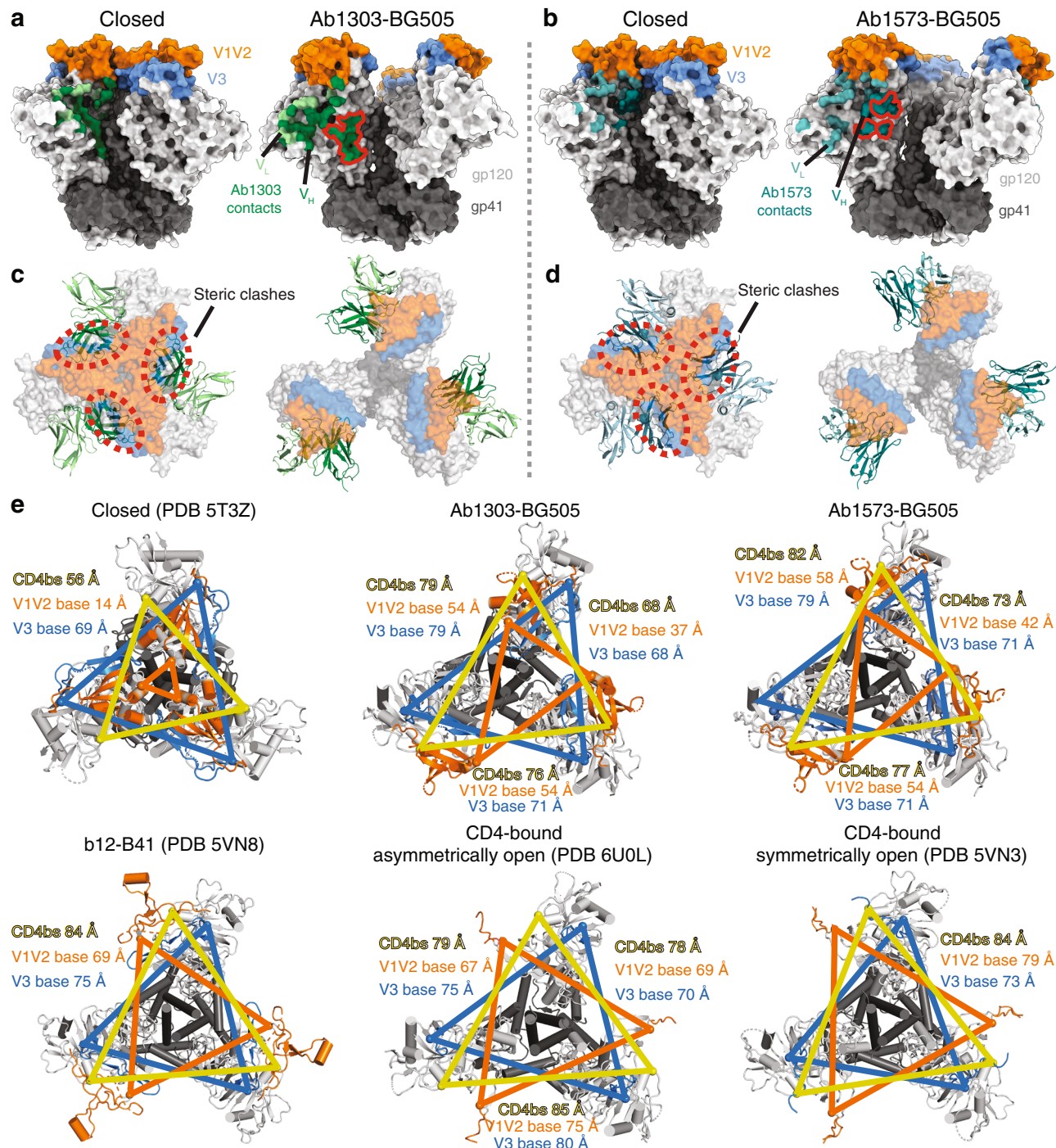

**Fig. 5 Ab1303 and Ab1573 bind an Env trimeric state distinct from the closed and CD4-bound open trimer conformations. a,b** Ab1303 (**a**) or Ab1573 (**b**) contacts mapped onto a side-view surface representation of a closed BG505 trimer (PDB code 5CEZ) (left) and the occluded-open trimer to which each mAb binds (right). Regions of the antibody epitope that are buried in the closed state but accessible in the Ab1303- or Ab1573-bound state are outlined in red on the occluded-open Env trimer structures. **c, d** Cartoon representations of Ab1303 (**c**) or Ab1573 (**d**) interacting with closed (left) or occluded-open (right) Env trimers (seen from the top). Regions with steric clashes between the Fab and the closed trimer indicated by dashed, red ovals. **e** Comparison of inter-protomer distances of Cα atoms from selected residues at the CD4bs (yellow), the V1V2 base (orange), and the V3 base (blue) in a closed Env trimer (PDB 5T3Z), Ab1303-bound BG505 trimer, Ab1573-bound BG505 trimer, b12-bound B41 trimer (PDB 5VN8), a CD4-bound asymmetrically-open BG505 trimer (PDB 6U0L), and a sCD4-bound symmetrically-open B41 trimer (PDB 5VN3).

distance in a closed BG505 Env structure (Fig. 7b, c; red vertical line). The other major peaks for the unliganded BG505 sample, including major peaks at distances between ~20 Å and ~35 Å, were not readily interpretable based on closed SOSIP Env structures. However, the presence of the inter-protomer distances

other than 38 Å suggests that the unliganded BG505 SOSIP trimer can adopt conformational states in addition to the known closed, prefusion conformation. The broad heterogeneity of conformational states seen here may have been induced by incubation at 37 °C.

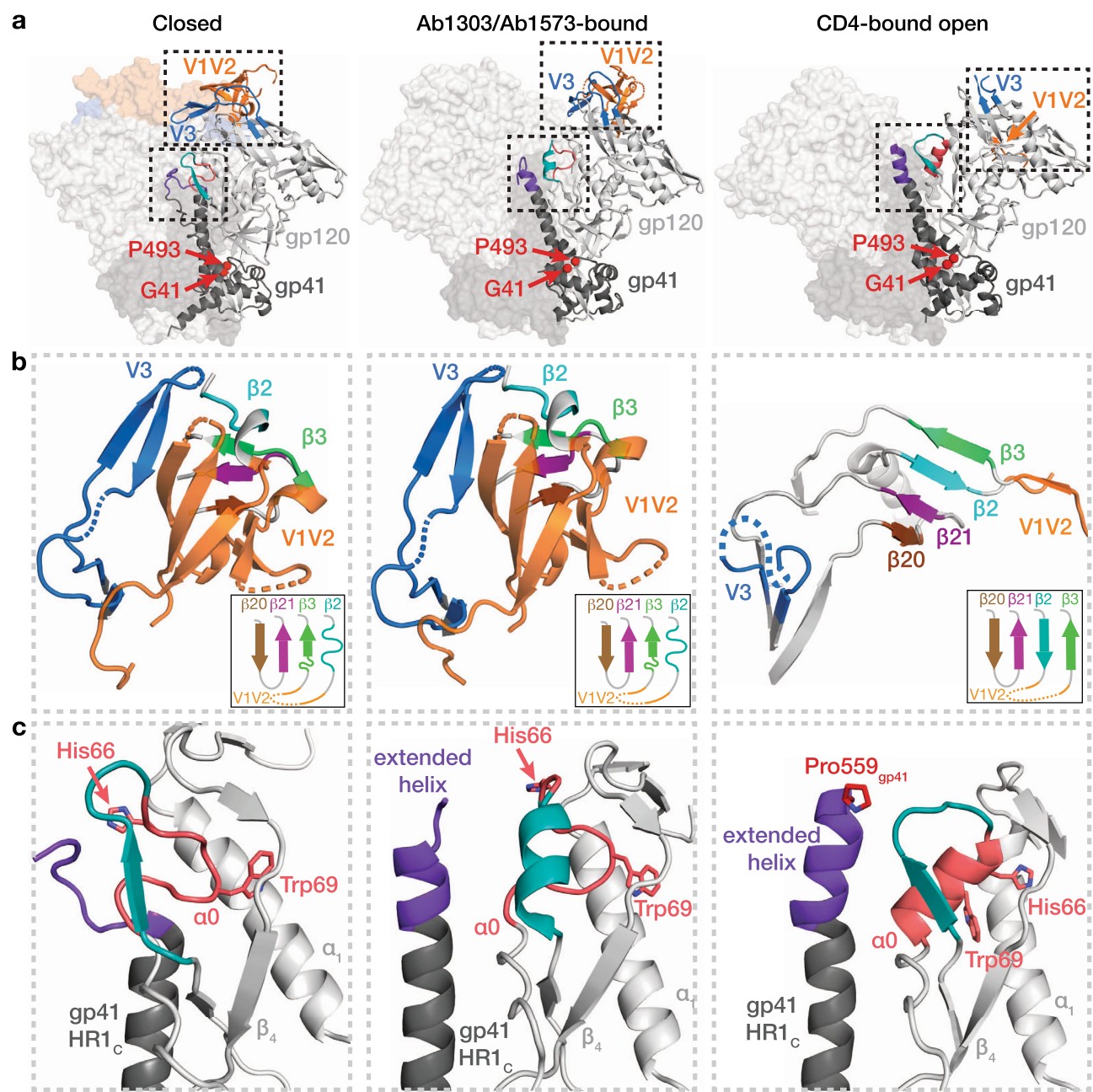

**Fig. 6 Structural changes between Env trimer states. a** Conformations of gp120/gp41 protomers in closed (left), Ab1303/Ab1573-bound (middle), and CD4-bound (right) states. The gp120 V1V2-V3 regions and gp120 α0/gp41 HR1C regions are highlighted in dashed boxes. Positions of $Gly41_{gp120}$ and $Pro493_{gp120}$ are shown as red spheres. **b** Structural rearrangements of gp120 V1V2 and V3 regions in closed (left), Ab1303/Ab1573-bound (middle), and CD4-bound (right) states. Secondary structures and relative locations of the 3-stranded sheet (left and middle) and rearranged 4-stranded bridging sheet (right) are highlighted, and respective topology models are shown in insets. Dotted lines represent disordered regions. **c** Comparison of gp120 α0 and neighboring region in closed (left), Ab1303/Ab1573-bound (middle), and CD4-bound (right) structures. Residues $Asp57_{gp120}$ - $Glu62_{gp120}$ are teal, α0 residues ($Lys65_{gp120}$ - $Ala73_{gp120}$) are salmon, with the $His66_{gp120}$ and $Trp69_{gp120}$ side chains shown. The N-terminal segment of gp41 $HR1_C$ is dark purple.

We also collected DEER data for BG505 complexes with Ab1303 and Ab1573 (green and cyan traces in Fig. 7b, c, respectively). Some of the short distance peaks, most notably a single major peak at ~24 Å, were observed for both antibody-bound Envs (Fig. 7b, c). In addition to this structurally uninterpretable peak, also present in both spectra were peaks at ~43 Å and ~55 Å, likely corresponding to the structurally-measured inter-protomer distance of 40 Å/46 Å (measured distance 1 for Ab1303-Env and Ab1573 Env complexes) and a combination of the 53 Å/58 Å (Ab1303) and 53 Å/60 Å (Ab1573) distances (measured distances 2 and 3; green vertical lines in Fig. 7b; cyan vertical lines in Fig. 7c). Peaks at or close to

these distances were found in the unliganded BG505 DEER spectrum, suggesting that the conformational states observed for Ab1303/Ab1573-binding Env also exist at a lower population in unliganded BG505. Interestingly, peaks near 67 Å—the measured inter-protomer distance for residue 174 in a b12-bound Env trimer —are observed in both the Ab1303-bound and Ab1573-bound Envs (major peak in Ab1303-bound Env spectrum; a minor peak in the Ab1573-bound spectrum), suggesting that binding of these antibodies induced a sub-population of Envs with a b12-bound conformation that was not captured in the cryo-EM structures.

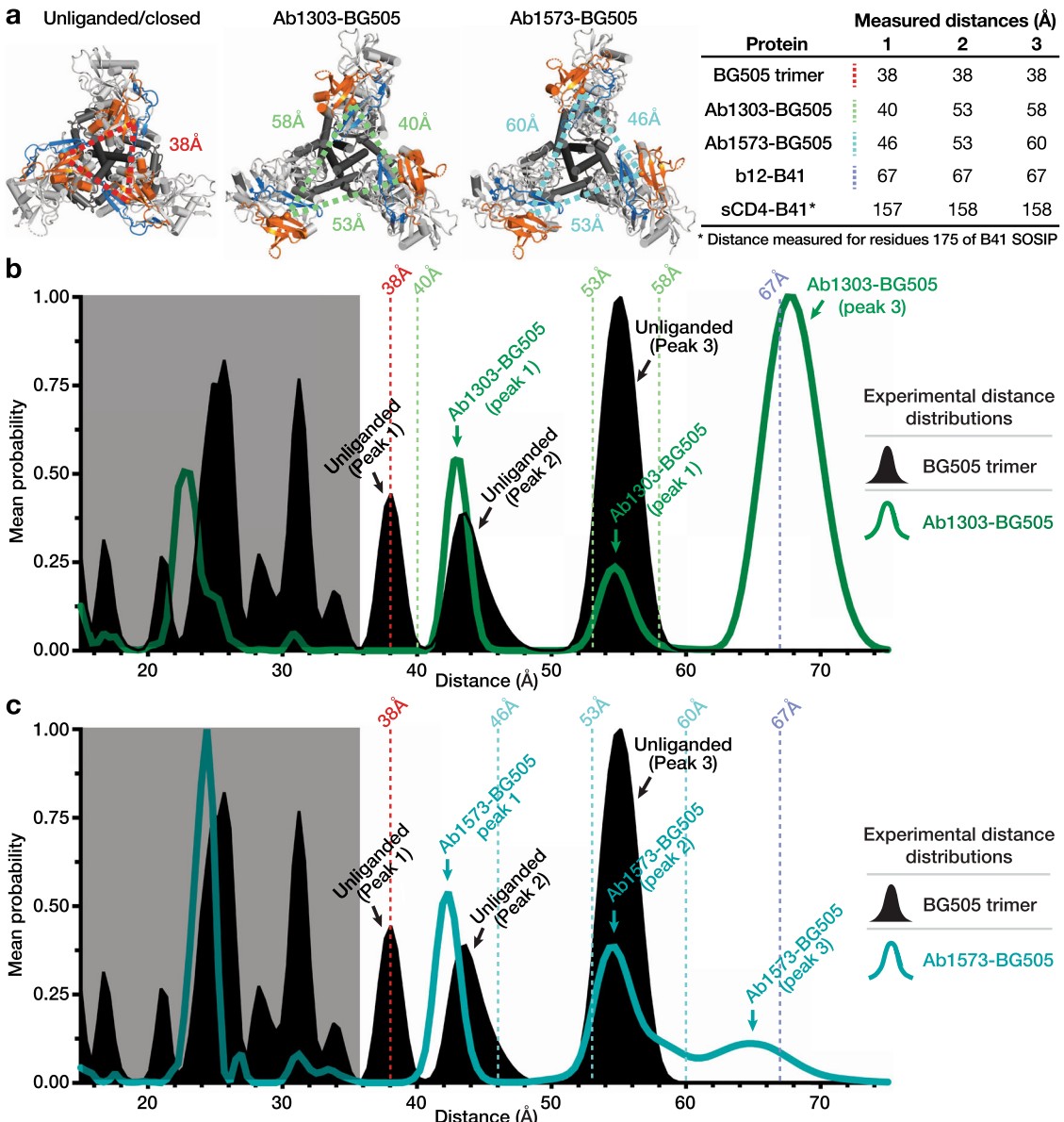

**Fig. 7 DEER spectra of unliganded BG505 trimer compared to Ab1303-bound and Ab1573-bound BG505 trimer. a** Left: Triangles showing distances between residue 174 Cα atoms on cartoon representations of Env trimers seen from the top. Right: Inter-protomer distances between Cα atoms for V1V2 residue 174 in different Env conformational states measured using coordinates from the indicated structures (BG505 trimer: PDB 5T3Z; Ab1303-BG505 and Ab-1573-BG505: this study; b12-B41: PDB 5VN8; CD4-B41: PDB 5VN3). **b, c** Left: Distance distributions for spin labels attached to V1V2 residue 174 in unliganded BG505 Env (solid black), Ab1303-bound BG505 (green trace in b), and Ab1573-bound BG505 (teal trace in c). Vertical dashed lines indicate the inter-subunit distances measured from coordinates (**a**) for each site in structures of closed, unliganded Env (red line), Ab1303-bound Env (green dashed lines in **b**), Ab-1573-bound Env (teal dashed lines in **c**), and b12-bound Env (purple dashed line). Right: Legend for DEER spectra. The distance distributions smaller than 38 Å were boxed in light gray shades. Source data are provided as a Source Data file.

## Discussion

Many HIV-1 vaccine efforts focus on using soluble Env trimers as immunogens to raise bNAbs[43]. Here we characterize Ab1303 and Ab1573, two CD4bs bNAbs raised by sequential immunization in NHPs of SOSIP trimer immunogens attached to VLPs[19]. Unexpectedly, both antibodies bind the CD4bs of an open form of HIV-1 Env trimer rather than the closed, prefusion state typically targeted by bNAbs raised in humans by natural infection[10]. In common with CD4-bound Envs, the trimer conformation recognized by Ab1303 and Ab1573 includes outward rotations of gp120, but the gp120 V1V2 loops are not rearranged to expose the coreceptor binding site on V3; thus, the Env trimer is open but the co-receptor binding site is occluded. The Ab1303/

Ab1573-bound occluded-open Env trimer conformation shares structural features with the conformation recognized by b12[11], an early CD4bs bNAb isolated from a phage display library[15]. Of relevance to immunogen design efforts is whether the occluded-open Env conformation exposes new epitopes that might elicit off-target non-neutralizing antibodies against trimer surfaces that would be buried in closed, prefusion Env trimers. Comparisons of BSAs between closed and occluded-open Env structures show that regions of V1V2 that are inaccessible in closed trimers (purple in Supplementary Fig. 4) might be accessible for antibody binding in the occluded-open Env conformation.

The question of which features of an Env-binding ligand induce an open coreceptor-binding HIV-1 Env conformation is prompted

by the existence of two distinct open trimer conformations (Fig. 6a): (i) the CD4-bound open trimer, a coreceptor-binding conformation in which V1V2 relocates to the sides of the Env trimer to expose V3 and form a 4-stranded gp120 bridging β-sheet[11–14], vs (ii) the occluded-open trimer in which the gp120s rotate outwards, but V1V2 remains "on top" of gp120 to shield the coreceptor binding site. Structures of the b12-Env[11] and the Ab1303/Ab1573-Env complexes reported here demonstrate that Env opening through gp120 rotation is not sufficient to induce the further structural rearrangements associated with CD4 binding (Fig. 6b, c). One difference that distinguishes CD4 from b12, Ab1303, Ab1573 and most other CD4bs bNAbs is that the antibodies lack a counterpart of CD4 residue Phe43, which inserts into the "Phe43" pocket on gp120[44]. We showed that small molecule CD4 mimetic entry inhibitors that insert into the gp120 Phe43 pocket recognize the CD4-bound open trimer conformation[35], whereas CD4 mimetics drugs that bind orthogonally to the Phe43 pocket bind closed, prefusion Envs[45]. Interaction with the gp120 Phe43 pocket may be necessary for recognition of the CD4-bound open trimer conformation, but is unlikely to be sufficient since CD4bs bNAbs such as N6[46] contain a CD4 Phe43 counterpart within their CDRH2 region, yet bind closed, prefusion Env trimers[47].

Our findings suggest that portions of the Ab1303 and Ab1573 epitopes on gp120 are buried on a closed, prefusion Env trimer (Fig. 5a) and that there are potential steric clashes between the Fabs and Env when they are docked onto their respective binding sites of closed Env (Fig. 5c, d). This implies that the Env trimer conformation that triggers development of this type of CD4bs bNAb is similar or equivalent to the occluded-open Env conformation described here. Indeed, DEER spectroscopy experiments suggested that a population of unliganded BG505 SOSIP Envs that had been incubated at 37 °C adopted a conformation consistent with the occluded-open conformation recognized by Ab1303 and Ab1573 (Fig. 7), therefore this conformation may have been present on at least a subset of the SOSIP-based immunogens used in the sequentially-immunized NHPs from which these antibodies were derived[19]. In addition, the Env trimers of HIV-1 strains that are neutralized by Ab1303 and Ab1573[19] may more readily adopt the occluded-open conformation than Envs in neutralization-resistant strains.

The DEER results, together with the demonstration of temperature-dependent changes in Ab1303 and Ab1573 binding stoichiometry, suggest that physiological temperature facilitates conformational changes in soluble Env trimers that result in the occluded-open state. Our previous DEER studies to probe conformations of Env SOSIPs conducted with 4 °C incubations concluded that unliganded SOSIPs showed conformations that were consistent with the closed pre-fusion trimer conformation, with no evidence for the CD4- or b12-bound open states[38]. For example, DEER spectra of the unliganded BG505 SOSIP labeled in V1V2 (residue 173) prepared at 4 °C showed a dominant inter-protomer distance signal between 30 and 40 Å, consistent with distances measured for the closed Env conformation[38]. In this study, the comparable unliganded BG505 SOSIP sample labeled in V1V2 (residue 174) prepared at 37 °C reported interspin distances consistent with the Ab1303/Ab1573-bound occluded-open trimer conformation. This suggests that at 37 °C, the temperature at which antibodies are generated in vivo, Env trimers attached to VLPs can adopt different conformational states between defined closed and open conformations. Whether the occluded-open conformation is present on membrane-bound viral Env trimers remains unknown, but the isolation of the b12 bNAb from a phage display library constructed using bone marrow from an HIV-1–infected individual is consistent with the idea that viruses include Envs with this or a similar conformation. The discovery that the b12-bound conformation of HIV-1 Env trimer is recognized by vaccination-induced neutralizing antibodies suggests the occluded-open conformation of HIV-1 as a potential target for immunogen design.

## Methods

**Protein expression and purification.** The native-like, soluble HIV Env gp140 trimer BG505 SOSIP.664 construct with 'SOS' mutations (A501C$_{gp120}$, T605C$_{gp41}$), the 'IP' mutation (I559P$_{gp41}$), N-linked glycosylation site mutation (T332N$_{gp120}$), an enhanced furin protease cleavage site (REKR to RRRRRR), and truncation after the C-terminus of gp41 residue 664[9] was cloned into pTT5 vector (National Research Council of Canada) and expressed in transiently-transfected Expi293F cells. For DEER experiments involving nitroxide spin labeling, BG505 SOSIP.664 was modified to include a free cysteine at residue Ser174$_{gp120}$ (S174C) by site-directed mutagenesis as described[38]. BG505 and BG505 S174C mutant Env trimers were purified from transfected cell supernatants as described[48] by 2G12 immunoaffinity and SEC with a HiLoad 16/600 Superdex 200 pg column followed by Superose 6 Increase 10/300 GL column (Cytiva).

The heavy and light chains of 6x-His tagged Ab1303 and Ab1573 Fabs were expressed in transiently-transfected Expi293F cells and purified by Ni-NTA chromatography followed by SEC as described[19]. IgG proteins were expressed in transiently-transfected Expi293F cells and purified by protein A affinity chromatography (Cytiva) followed by SEC as described[21,48].

**Competition ELISA.** BG505 SOSIP trimers were randomly biotinylated using the EZ-Link NHS-PEG4-Biotin kit (Thermo Fisher Scientific) according to the manufacturer's guidelines. Based on the Pierce Biotin Quantitation kit (Thermo Fisher Scientific), the number of biotin molecules per protomer was estimated to be 1.5. Biotinylated BG505 SOSIP timers were immobilized on Streptavidin-coated 96-well plates (Thermo Fisher Scientific) at a concentration of 5 µg/mL in blocking buffer (1% BSA + 1% goat serum in TBS-T) for 1 h at room temperature (RT). After washing plates once in TBS-T, plates were incubated with a Fab derived from a bNAb that targets the V3 loop (10-1074), the fusion peptide (VRC34), the V1V2 loop (PG16), or the CD4bs (3BNC117), at a concentration of 100 µg/mL in blocking buffer for 1 h at RT. After washing plates twice in TBS-T, a concentration series of Ab1303 or Ab1573 IgG was added to the Fab-BG505 complexes with a top concentration of 100 µg/mL in blocking buffer and fourfold dilutions for 1 h at 37 °C. After washing plates three times in TBS-T, bound IgG was detected using an HRP-conjugated goat anti-human Fc antibody (Southern Biotech) at a dilution of 1:10,000 in blocking buffer for 1 h at RT. After washing plates three times with TBS-T, 1-Step Ultra TMB substrate (Thermo Fisher Scientific) was added for 5 min and plates were analyzed using a plate reader (BioTek).

**PGT145 capture ELISA.** This capture ELISA was performed as described previously with minor modifications[9]. Briefly, Corning Costar 96-Well Assay high binding plates (07-200-39) were coated and incubated overnight at 4 °C with PGT145 IgG at 5 µg/ml in 0.1 M NaHCO$_3$ (pH 9.6). Unbound PGT145 IgG was removed, and wells were blocked with 3% BSA in TBS-T (20 mM Tris, 150 mM NaCl, 0.1% Tween20) for 1 h at RT. BG505 SOSIP.664 was added at 10 µg/ml and incubated for 1 h at RT, then removed. Serially diluted Fabs in 3% bovine serum albumin in TBS-T were added, incubated at RT for 2 h, then washed three times with TBS-T. Horseradish peroxidase labeled mouse anti-His tag antibody (GenScript: A00186) was added for 30 min at 1:1000 dilution, followed by three washes with TBS-T. 1-Step™ Ultra TMB-ELISA Substrate Solution (Thermo Fisher Scientific: 34029) was added for colorimetric detection. Color development was quenched with 1.0 N HCl, and absorption was measure at 450 nm. Two independent, biological replicates were performed.

**Multiangle light scattering coupled with size-exclusion chromatography.** Protein mixtures were characterized by size-exclusion chromatography/multiangle light scattering (SEC-MALS) on a Superose 6 10/300 GL column. The column was connected in-line with a light scattering detector (DAWN HELEOS II; Wyatt Technology), a dynamic light-scattering detector (DynaPro Nanostar; Wyatt Technology), and a refractive index detector (Optilab t-rEX; Wyatt Technology). Data were collected at RT with a flow rate of 0.5 mL/min. Data were analyzed and molecular mass estimations were generated using ASTRA 6 software (Wyatt Technology).

**X-ray crystallography.** Crystallization screens for Ab1303 Fab and Ab1573 Fab were performed using the sitting drop vapor diffusion method at RT by mixing concentrated Fabs with an equal amount of reservoir solution (Hampton Research) using a TTP Labtech Mosquito automatic microliter pipetting robot. Ab1303 Fab crystals were obtained in 10% (v/v) PEG 200, 0.1 M Bis-Tris propane (pH 9.0), and 18% w/v PEG 8,000. Ab1573 Fab crystals were obtained in 0.1 M Tris (pH 8.2), 26% (w/v) PEG 4000. Ab1303 Fab crystals were directly looped and cryopreserved in liquid nitrogen, whereas Ab1573 Fab was briefly mixed with 15% glycerol cryoprotectant solution before cryopreservation in liquid nitrogen.

X-ray diffraction data were collected at Stanford Synchrotron Radiation Lightsource (SSRL) beamline 12-1 equipped with an Eiger X 16 M pixel detector

(Dectris) at a wavelength of 0.97946 Å. Recorded data were indexed, integrated, scaled in XDS[49,50] and merged with AIMLESS v0.7.4[51]. The structure of Ab1303 Fab was determined by molecular replacement using PHASER v2.8.2[52] and a search model comprising separate $V_H$-$V_L$ and $C_H1$-$C_L$ domains of a human antibody (PDB 4YK4) with the CDR loops removed. The structure of Ab1573 Fab was determined similarly, except using the Ab1303 Fab as the search model. Coordinates of both Fabs were refined using Phenix v1.19.2[53,54] and iterations of manual building in Coot v0.9[55] (Supplementary Table 2).

**Cryo-EM sample preparation.** Ab1303-BG505 and Ab1573-BG505 complexes were prepared by incubating purified and concentrated Ab1303 and Ab1573 Fabs with BG505 SOSIP.664 trimer at a molar ratio of (3.6:1 Fab:Env) at 37 °C for 2 h. A final concentration of 0.05% (w/v) fluorinated octylmaltoside (Anatrace) was added to both samples immediately before cryo-freezing. Cryo-EM grids were prepared using a Mark IV Vitrobot (Thermo Fisher) operated at 12 °C and 95% humidity. 2.6 μL of concentrated sample was applied to 300 mesh Quantifoil R1.2/1.3 grids, blotted for 4 s, and grids were then vitrified in liquid ethane. The 12 °C temperature of the Vitrobot reduced ice on the grids and prevented fog building up in the chamber. For sample freezing, multiple pairs of tweezers preheated to 37 °C before picking up a new grid. The time between loading a sample drop onto a grid and plunge freezing was only a few seconds. Thus, the 37 °C temperature of the Fab:Env incubation was likely to have been maintained during the few seconds in the 12 °C Vitrobot.

**Cryo-EM data collection and processing.** Cryo-grids were loaded onto a 200 kV Talos Arctica electron microscope (Thermo Fisher) (Ab1303-BG505 complex) or a 300 kV Titan Krios electron microscope (Thermo Fisher) equipped with a GIF Quantum energy filter (slit width 20 eV) operating at a nominal ×105,000 magnification (Ab1573-BG505). For data collection on the Krios, defocus ranges for both Ab1303-BG505 and Ab1573-BG505 datasets were set to 1.4–3.0 μm. Movies were recorded using a 6k × 4k Gatan K3 direct electron detector operating in super-resolution mode with a pixel size of 0.869 Å•pixel⁻¹ (Arctica) or 0.855 Å•pixel⁻¹ (Krios) and collected using SerialEM v3.7 software. The recorded movies were sectioned into 40 subframes with a total dose of 60 e⁻•Å⁻², generating a dose rate of 1.5 e⁻/Å²•subframe. A total of 8036 (Ab1303-BG505) and 8478 (Ab1573-BG505) movies were motion-corrected using MotionCor2[56] with 2× binning. The CTFs of motion-corrected micrographs were estimated using CTFFIND v4.1.14[57]. For both datasets, a set of ~1000 particles were manually picked and reference-free 2D classes were selected for automatic particle picking using RELION AutoPicking[58,59]. Automatically picked particles were subjected to iterations of reference-free 2D class averaging. A closed-conformation trimer (PDB 5CEZ) map that was low-pass filtered to 80 Å was used as reference for 3D classifications and high-resolution 3D refinement in RELION v3.1[58,59]. CTF refinements were performed on particles used previously in 3D refinement, and the CTF-refined particles were subsequently polished and subjected to a last iteration of 3D refinement and map sharpening. 3D FSCs of maps were calculated using the Remote 3DFSC Processing Server as described[60]. Local resolutions of the refined maps were calculated using RELION v3.1[58,59].

**Model building.** Coordinates for gp120, gp41, Ab1303 Fab, and Ab1573 Fab were fitted into the corresponding regions of the density maps. The following coordinate files were used for initial fitting: BG505 gp120 monomer (PDB 5T3Z), gp41 monomer (PDB 5T3Z), and crystal structures of Ab1303 and Ab1573 (this study). Coordinates for the two Fab-BG505 structures and N-linked glycans were manually refined and built in Coot[55]. Iterations of whole-complex refinements using phenix.real_space_refine[53,54] and manual refinements were performed to correct for interatomic bonds and angles, clashes, residue side chain rotamers, and residue Ramachandran outliers.

**Structural analyses.** CDR lengths were derived based on IMGT definitions[61]. Structural figures were made using PyMOL v2.3 (Schrödinger, LLC) or ChimeraX v0.9[62]. Buried surface areas (BSAs) were calculated using PDBePISA[63] and a 1.4 Å probe. Potential hydrogen bonds were assigned using the geometry criteria with separation distance of <3.5 Å and A-D-H angle of >90°. The maximum distance allowed for a potential van der Waals interaction was 4.0 Å. Protein surface electrostatic potentials were calculated in PyMOL v2.3 (Schrödinger LLC). Briefly, hydrogens were added to proteins using PDB2PQR[64], and an electrostatic potential map was calculated using APBS[65]. Epitopes for antibodies in Fig. 3 were identified as gp120 residues containing an atom within 4 Å of an antibody as calculated in PyMOL v2.3 (Schrödinger, LLC).

**SOSIP V1 spin labeling and pulsed DEER spectroscopy.** SOSIP spin labeling and pulsed DEER spectroscopy were performed similarly to methods described previously[38]. Briefly, purified BG505 S174C SOSIP protein was concentrated to ~100 μM in TBS (pH 7.4) and reduced with tris(2-carboxyethyl)phosphine (TCEP) buffer such that the final concentration of TCEP was in a 2× molar excess relative to the target cysteine residue for 1 h. TCEP was removed using a desalting column (Zeba, 89883) and the reduced protein was then incubated with five-molar excess of the V1 nitroxide spin label (bis(2,2,5,5-tetramethyl-3- imidazoline-1-oxyl-4-il)-

disulfide) for 5 h at RT then at 4 °C overnight. A SEC column (Superose 6 Increase 10/300 GL) was used to remove excess V1 spin label. V1-labeled SOSIP was then buffer exchanged into deuterated solvent containing 20% glycerol. Unliganded V1-labeled SOSIP and V1-labeled SOSIP incubated with a three-molar excess of Ab1303 Fab or Ab1573 Fab were placed at 37 °C for 3 h immediately prior to flash freezing.

For DEER spectroscopy, approximately 60 μL samples of ~150 μM spin-labeled protein complexes were flash frozen within a 2.0/2.4 mm borosilicate capillary (Vitrocom, Mountain Lakes, NJ) in liquid nitrogen. Sample temperature was maintained at 50 K during data collection by a recirculating/closed-loop helium cryocooler and compressor system (Cold Edge Technologies, Allentown, PA). Four-pulse DEER spectroscopy data were collected on a Q-band Bruker ELEXSYS 580 spectrometer using a 150 W TWT amplifier (Applied Engineering Systems, Fort Worth, TX) and an E5106400 cavity resonator (Bruker Biospin). Pulse lengths were optimized via nutation experiment but ranged from 21 to 22 ns (π/2) and 42 to 44 ns (π); Observer frequency was set to a spectral position 2 G downfield of the low and central resonance intersection minimum in the absorption spectrum, and the pump envelope frequency was a 50 MHz half-width square-chirp pulse (generated by a Bruker arbitrary waveform generator) set 70 MHz downfield from the observer frequency. Dipolar data were analyzed using LongDistances v.932, a custom program written by Christian Altenbach in LabVIEW (National Instruments); software available online (http://www.biochemistry.ucla.edu/biochem/Faculty/Hubbell/) and described elsewhere[66]. As demonstrated previously, control experiments in which mock-labeled native SOSIPs without an introduced unpaired cysteine did not exhibit DEER signals above background[38].

**Reporting summary.** Further information on research design is available in the Nature Research Reporting Summary linked to this article.

## Data availability

The atomic models generated in this study have been deposited in the Protein Data Bank (PDB) under accession codes 7RYU and 7RYV for Ab1303 Fab and Ab1573 Fab X-ray crystal structures, respectively. The cryo-EM maps and atomic coordinates have been deposited in the Electron Microscopy Data Bank (EMDB) and Protein Data Bank with accession codes EMD-25877 and PDB 7TFN for Ab1303-BG505 complex, and EMD-25878 and PDB 7TFO for Ab1573-BG505 complex. Source data are provided with this paper.

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

## Acknowledgements

We thank Anthony P. West (Caltech) for help with analysis of antibody sequences. Cryo-EM was performed in the Beckman Institute Resource Center for Transmission Electron Microscopy at Caltech with assistance from directors A. Malyutin and S. Chen. We thank the Gordon and Betty Moore and Beckman Foundations for gifts to Caltech to support the Molecular Observatory (Dr. Jens Kaiser, director) and the beamline staff at SSRL for data collection assistance. Use of the Stanford Synchrotron Radiation Lightsource, SLAC National Accelerator Laboratory, is supported by the U.S. Department of Energy, Office of Science, Office of Basic Energy Sciences under Contract No. DE-AC02-c76SF00515. The SSRL Structural Molecular Biology Program is supported by the DOE Office of Biological and Environmental Research, and by the National Institutes of Health, National Institute of General Medical Sciences (P41GM103393). The contents of this publication are solely the responsibility of the authors and do not necessarily represent the official views of NIGMS or NIH. We thank Jost Vielmetter and Pauline Hoffman at the Beckman Institute Protein Expression Center at Caltech for protein production, John Moore (Weill Cornell Medical College) for the BG505 stable cell line, Kristie M. Gordon (The Rockefeller University) for assistance with flow cytometry, and Rogier W. Sanders and Marit J. van Gils (Academisch Medisch Centrum Universiteit van Amsterdam) for providing AviTagged and biotinylated BG505 and B41 SOSIP trimers for sorting. This work was supported by the National Institute of Allergy and Infectious Diseases (NIAID) HIVRAD P01 AI100148 (to P.J.B. and M.C.N.), Gates CAVD grant INV-002143 (to P.J.B., M.A.M., and M.C.N.), the Intramural Research Program of the National Institute of Allergy and Infectious Diseases, NIH. (R.G and M.A.M), NIH P50 AI150464 (P.J.B.). M.C.N. is an HHMI Investigator.

## Author contributions

Z.Y., K.A.D., M.A.M., M.C.N., W.L.H., and P.J.B. designed the research. Z.Y., K.A.D., M.D.B., M.A.G.H., H.B.G., A.T.D, A.E., and R.G. performed experiments and analyzed results. Z.Y., K.A.D., M.D.B., and P.J.B. wrote the paper with input from co-authors.

## Competing interests

The authors declare no competing interests.

## Additional information

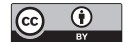

