## [Peer Review File · Nature Communications]

Reviewer comments, first round review:

Reviewer #1 (Remarks to the Author):

Yang et al. have characterized two CD4-binding site (CD4bs) neutralizing antibodies, Ab1303 and Ab1573, isolated by single cell cloning from SOSIP-binding B cells derived from sequentially-immunized macaques. The epitopes of these antibodies were first mapped to the CD4 bs by competition ELISA, and they also did not bind the PGT145-captured SOSIP trimer, suggesting that the antibodies recognize a conformation of the Env trimer different from the closed prefusion state. The authors then determined the Fab-Env cryo-EM structures, showing that both antibodies indeed recognize the CD4bs on the Env trimer with an 'occluded-open' conformation, with gp120 subunits outwardly-rotated, but no V1V2 displacement, co-receptor binding site exposure, or formation of the bridging sheet. Additional experiments by double electron-electron resonance (DEER) spectroscopy suggested an equilibrium between occluded-open and closed Env conformations in solution. The newly identified open conformation of HIV-1 Env trimer, accessible by immunization, may inform immunogen design and antibody therapeutic efforts.

This is a very detailed and comprehensive study of two new monoclonal antibodies identified from the macaques immunized by some promising vaccine candidates. Numerous anti-CD4 bs antibodies have been reported previously, but those isolated by B cell cloning from vaccinated animals are rare and they can guide immunogen optimization. The manuscript is well written and all data seem technically sound. It should therefore be of interest to the readers in the HIV vaccine field. Several specific comments should be addressed to improve the manuscript.

1. Apparently, there is another manuscript by the authors describing neutralization properties of these antibodies that has been submitted (ref 19), but the results are briefly summarized here - "Ab1303 neutralized 12 of the 19 cross-clade strain panel with IC50 values <100 µg/mL, whereas Ab1573 neutralized five strains in the panel with IC50 values <100 µg/mL19". Although definition of neutralization is somewhat arbitrary, the widely accepted cutoff in a standard pseudovirus assay is 50 µg/mL for IC50 values. Imagine that others may start calling their antibodies neutralizing by redefining the cutoff value, say, to 500 µg/mL, then they could claim that many existing vaccine candidates have already induced bnAbs, which of course is meaningless. Nevertheless, it would be important to report how good these antibodies are in potency and breadth, as compared to those so-called strain-specific neutralizing or non-neutralizing CD4 bs antibodies, such as b6.
2. The SOSIP Env trimers have been used throughout the manuscript (binding, cryo-EM and DEER) and they are designed with artificial modifications to stabilize the prefusion closed conformation. The caveat of using these trimers should be discussed since the "occluded-open" conformation may not be stable or even not exist in the context of the unmodified Env trimer on the virion surface. It may be relevant to the SOSIP-based immunogen design, but perhaps less so to our understanding of the viral entry process.
3. During the sample preparation for the DEER spectroscopy, "purified SOSIP proteins were concentrated to ~100 µM in TBS 574 (pH 7.4) and reduced with tris(2-carboxyethyl)phosphine (TCEP) buffer such that the final concentration of TCEP was in a 2x molar excess relative to the target cysteine residue for one hour. TCEP was removed using a desalting column and the reduced protein was then incubated with 5 molar excess of the V1 nitroxide spin label (bis(2,2,5,5-tetramethyl-3- 578 imidazoline-1-oxyl-4-il)-disulfide)". It is unclear whether some native disulfides in the Env trimer could also be reduced by TCEP and subsequently labelled. In addition, it does not seem that the labelled protein has been characterized at least by the antibody binding assays to be sure there are no unwanted structural changes. After all, there are "The other major peaks for the unliganded BG505 sample, including major peaks at distances between ~20Å and ~35Å, were not readily interpretable based on closed SOSIP Env structures". Could some of the peaks result from the sample preparation?
4. Line 49, "Coreceptor binding to gp120 results in further conformational changes including insertion of the gp41 fusion peptide into the host cell membrane". Ref 6 is a review article, what is the evidence for further conformational changes in gp120 induced by coreceptor binding?

Reviewer comments, first round review:

Reviewer #2 (Remarks to the Author):

The impact of this manuscript is understanding the biophysical properties of efficacious anti-HIV Abs produced through sophisticated immunization protocols. Overall, these are important, informative results - though more so for the specializing community. The experiments are rigorously performed and interpreted, completely described, and quite accessibly presented in both the text and figures. The synergistic combination of multiple sophisticated structural techniques is quite pleasing and generates confidence in the conclusions. The only minor ding is that the final refinement statistics for the high-resolution Fab structure aren't the best. However, the manuscript is worthy of publication as-is.

Reviewer #3 (Remarks to the Author):

In this manuscript the authors structurally characterize two CD4-binding site (CD4bs) antibodies, isolated from sequentially immunized macaques. Both antibodies exhibited weak, heterologous neutralization. The antibodies recognized a partially open or "occluded" conformation, characterized by cryo-EM. By using DEER spectroscopy, the authors show evidence that an equilibrium between occluded-open and closed Env conformations, suggesting that Ab1303/Ab1573 binding stabilizes an existing conformation.

This study provides useful information on the immune response to commonly used HIV-1 vaccine immunogens, specifically on CD4bs Abs that target partially open Env conformations. The study also provides insights into the conformational heterogeneity of Env immunogens under physiological conditions. The combined use cryo-EM and DEER spectroscopy shows that the antibodies AB1302 and AB1573 bind a conformation of Env that exists is sampled within unliganded populations of these Env immunogens. That these antibodies neutralize HIV-1, albeit weakly, suggests that this Env conformation is also sampled on native virus. The paper is well-written and easy to read. The neutralization data for these antibodies has been submitted for review in a separate manuscript (reference #19).

Comments for authors:

1. Figure 1C shows SEC-MALS profiles of the antibody complexes at 25C and 37C, which forms the basis for the conclusion of altered stoichiometry. Given the importance of these conclusions in the context of this study, an orthogonal measure of stoichiometry, such as from negative stain electron microscopy (NSEM), a technique that should be easily accessible to this group, will make the results stronger. Aside from visualizing the stoichiometry of antibodies bound to Env, it will also be interesting to see if any differences are due to altered Env conformations at the different temperatures – these should be revealed even at the lower NSEM resolutions.
2. There are no methods associated with the SEC-MALS experiments and it is unclear what ratio of Env:antibody was used for these experiments.
3. The authors show temperature dependent differences in the antibody-bound complexes and show greater Env mobility at 37C. The cryo-EM samples were prepared by incubating at 37C for 2 hrs, although later the sample was exposed to lower temperature ~12C in the Vitrobot for grid plunging. Could the complex change between 37C and 12C? This seems like a relevant question for this study where temperature-dependent Env differences are observed. This can be determined relatively easily by incubating the antibody-Env complex first at 37, and then at 12C followed by assessing stoichiometry by SEC-MALS and NSEM.
4. At the end of the first Results para, the authors state: "Here we investigated Ab1303 and Ab1573, which unexpectedly recognized the CD4bs rather than the V3-glycan patch that was targeted in the sequential immunization scheme." It is unclear from the information provided why this result is "unexpected". Was the CD4 binding site in the immunogens blocked in some way to prevent antibody access? If not, it is not really unexpected that the immune system should elicit antibodies targeting the CD4 binding site in the immunogen.
5. It is interesting that the two antibodies described in this study are not as mutated as broadly neutralizing CD4bs antibodies (such as VRC01 and others). Could this be due to reduced structural barriers for antibody access in the partially open Env conformation recognized by these antibodies? Perhaps the authors could comment on this in the discussion.
6. In the methods section "florinated octylmaltoside" should be "fluorinated octylmaltoside".
7. It is not clear from the figures and text how this antibody interacts with the CD4 binding loop. Would be good to add a figure showing interaction with this key Env region.

Reviewer comments, first round review:

8. The authors mention: "While the Ab1303 light chain was adjacent to the Asn276gp120 glycan, the Ab1573 heavy and light chains contacted the N-glycans attached to Asn197gp120, Asn276gp120, Asn363gp120, and Asn386gp120, likely resulting in glycan rearrangements to allow binding" It will be great to include a figure that shows the glycan rearrangements, by comparing to an Env structure that does not have an antibody bound in the vicinity of these glycans (such as PDB 5FYL).

Nov 28, 2021

Dear Reviewers,

Thank you for carefully reading our paper, **“Neutralizing antibodies induced in immunized macaques recognize the CD4-binding site on an occluded-open HIV-1 envelope trimer”** (NCOMMS-21-39911A). We have updated the manuscript to address the reviewers’ concerns, the updated sections can be found from the Word file’s “Comments”. We have responded to the reviewers’ concerns in our revised manuscript as follows:

Reviewer #1 (Remarks to the Author):

Yang et al. have characterized two CD4-binding site (CD4bs) neutralizing antibodies, Ab1303 and Ab1573, isolated by single cell cloning from SOSIP-binding B cells derived from sequentially-immunized macaques. The epitopes of these antibodies were first mapped to the CD4 bs by competition ELISA, and they also did not bind the PGT145-captured SOSIP trimer, suggesting that the antibodies recognize a conformation of the Env trimer different from the closed prefusion state. The authors then determined the Fab-Env cryo-EM structures, showing that both antibodies indeed recognize the CD4bs on the Env trimer with an ‘occluded-open’ conformation, with gp120 subunits outwardly-rotated, but no V1V2 displacement, co-receptor binding site exposure, or formation of the bridging sheet. Additional experiments by double electron-electron resonance (DEER) spectroscopy suggested an equilibrium between occluded-open and closed Env conformations in solution. The newly identified open conformation of HIV-1 Env trimer, accessible by immunization, may inform immunogen design and antibody therapeutic efforts.

This is a very detailed and comprehensive study of two new monoclonal antibodies identified from the macaques immunized by some promising vaccine candidates. Numerous anti-CD4 bs antibodies have been reported previously, but those isolated by B cell cloning from vaccinated animals are rare and they can guide immunogen optimization. The manuscript is well written and all data seem technically sound. It should therefore be of interest to the readers in the HIV vaccine field.

We thank the reviewer for their supportive comments about our manuscript.

Several specific comments should be addressed to improve the manuscript.

1. Apparently, there is another manuscript by the authors describing neutralization properties of these antibodies that has been submitted (ref 19), but the results are briefly summarized here - “Ab1303 neutralized 12 of the 19 cross-clade strain panel with IC50 values <100 µg/mL, whereas Ab1573 neutralized five strains in the panel with IC50 values <100 µg/mL 19”. Although definition of neutralization is somewhat arbitrary, the widely accepted cutoff in a standard pseudovirus assay is 50 µg/mL for IC50 values. Imagine that others may start calling their antibodies neutralizing by redefining the cutoff value, say, to 500 µg/mL, then they could claim that many existing vaccine candidates have already induced bnAbs, which of course is meaningless. Nevertheless, it would be important to report how good these antibodies are in potency and breadth, as compared to those so-called strain-specific neutralizing or non-neutralizing CD4 bs antibodies, such as b6.

The submitted paper we referred to was published on Nov 24, 2021 in *Science Translational Medicine* (<https://www.science.org/doi/10.1126/scitranslmed.abk1533>).

The reviewer raises a valid point regarding the neutralization IC₅₀ cutoff of 100 µg/mL in the *Science Translational Medicine* paper. We were interested in finding any hint of neutralization and so, at the direction of our co-author Michael Seaman (who is the director of the Gates Foundation CAVD neutralization center), we increased the starting concentration of Ab so we could detect neutralization at IC₅₀ values of 100 µg/mL.

2. The SOSIP Env trimers have been used throughout the manuscript (binding, cryo-EM and DEER) and they are designed with artificial modifications to stabilize the prefusion closed conformation. The caveat of using these trimers should be discussed since the “occluded-open” conformation may not be stable or even not exist in the context of the unmodified Env trimer on the virion surface. It may be relevant to the SOSIP-based immunogen design, but perhaps less so to our understanding of the viral entry process.

The b12-bound ‘occluded-open’ conformation was observed in Sriram Subramaniam’s cryo-ET/sub-tomogram averaging structures of HIV-1 Envs on virions (Liu et al., 2008, Nature). A higher-resolution single-particle cryo-EM structure of a b12-SOSIP Env complex (Ozorowski et al., 2017, Nature) fits well into the cryo-ET virion-bound b12-Env complex density, as do crystallographic and/or single-particle cryo-EM structures of closed, prefusion SOSIPs and CD4-bound open SOSIPs (see Figure 7 in Stadtmueller et al., 2018). In addition, a recent cryo-ET/subtomogram averaging study of virion-bound Env trimers reported trimer structures at higher resolutions (~9 Å) than the Liu/Subramaniam 2008 structures (~20 Å), concluding that virion-bound and SOSIP Env trimers adopt similar structures, including similar prefusion closed and CD4-induced open conformational states (Li et al., 2020).

In Stadtmueller et al., our previous DEER study of HIV-1 Env SOSIPs that included measurements of distances of unliganded Envs and Envs complexed with b12, other antibodies, and CD4, we showed that DEER measurements of SOSIPs were consistent with virion-bound Env structures. We also discussed evidence that Env trimers attached to virion membranes resemble SOSIP Env structures, including an analysis of how our DEER results differ from smFRET comparisons of SOSIP and virion-bound Env conformations. Thus, multiple lines of evidence suggest that conformational states for SOSIP Env trimer structures are relevant for virion-bound Env trimers.

3. During the sample preparation for the DEER spectroscopy, “purified SOSIP proteins were concentrated to ~100 µM in TBS 574 (pH 7.4) and reduced with tris(2-carboxyethyl)phosphine (TCEP) buffer such that the final concentration of TCEP was in a 2x molar excess relative to the target cysteine residue for one hour. TCEP was removed using a desalting column and the reduced protein was then incubated with 5 molar excess of the V1 nitroxide spin label (bis(2,2,5,5-tetramethyl-3-578 imidazoline-1-oxyl-4-yl)-disulfide)”. It is unclear whether some native disulfides in the Env trimer could also be reduced by TCEP and subsequently labelled.

Thank you for raising this important point. As now noted in the revised Methods section, in our previous DEER study (Stadtmueller et al., 2018) of the same BG505 SOSIP used for the current study, we described control experiments in which native SOSIPs without an introduced target cysteine underwent mock labeling. In those experiments, SOSIPs were reduced with TCEP and were incubated with the V1 nitroxide spin label. We reported that “...data for mock-labeled BG505 and B41 SOSIP proteins (purified proteins containing no introduced cysteines that were subjected to the V1 labeling procedure) did not exhibit signals above background...”

In addition, it does not seem that the labelled protein has been characterized at least by the antibody binding assays to be sure there are no unwanted structural changes. After all, there are “The other major peaks for the unliganded BG505 sample, including major peaks at

distances between ~20Å and ~35Å, were not readily interpretable based on closed SOSIP Env structures". Could some of the peaks result from the sample preparation?

In combination with our previous HIV-1 Env DEER study (Stadtmueller et al., 2018), our lab has shown that spin-labeled SOSIP Env trimers (both BG505 and B41) bind to a variety of antibodies, to soluble CD4, and to small molecule HIV-1 inhibitors. As we noted in the present paper (and more extensively discussed in Stadtmueller et al., 2018), "Although the V1 spin label is small (about the size of an amino acid) and contributes limited width to DEER distance distributions (Toledo Warshaviak et al., 2013), distances between spin label side chains measured by DEER only rarely equal the C α -C α inter-protomer distance since the radical center is found on the nitroxide ring, not the peptide linkage. As such, DEER results can be complicated by conformational heterogeneity and flexibility intrinsic to the protein studied." Thus, we do not expect to be able to interpret all distances derived from DEER measurements based on known structures.

Furthermore, in Stadtmueller et al., 2018, we modeled the V1 nitroxide spin label on DEER target sites on BG505 to understand how V1 rotamers might affect measured DEER distances (see Figure S1B). This modeling identified the number of and the most probable V1 rotamers for each target site, and we derived inter-protomer distances between rotamers. At most target sites, C α -C α distances were within 2-3 Å of measured V1 rotamer distances; however, one site showed considerable difference between C α -C α and V1 rotamer distances. At the target site in BG505 SOSIP in the present paper, V1 rotamers may affect DEER distance measurements. To reflect this, we added the following sentence to the text: "In addition, previous work to model V1 nitroxide side chain rotamers on BG505 Env DEER target sites suggested that differences in V1 rotamers can contribute to measured DEER distances³⁷."

We also updated Fig. 7 to distinguish distance distributions smaller and larger than the 38Å peak to make it more clear to the reader.

4. Line 49, "Coreceptor binding to gp120 results in further conformational changes including insertion of the gp41 fusion peptide into the host cell membrane". Ref 6 is a review article, what is the evidence for further conformational changes in gp120 induced by coreceptor binding?

All available CD4-bound HIV-1 Env structures (including both SOSIP and virion-bound Env structures) do not show enough exposure of the fusion peptide to allow insertion into the host cell membrane. We therefore assume, as suggested in the cited review article, that coreceptor binding causes further conformational changes that facilitate fusion peptide exposure and insertion into the host membrane.

Reviewer #2 (Remarks to the Author):

The impact of this manuscript is understanding the biophysical properties of efficacious anti-HIV Abs produced through sophisticated immunization protocols. Overall, these are important, informative results - though more so for the specializing community. The experiments are rigorously performed and interpreted, completely described, and quite accessibly presented in both the text and figures. The synergistic combination of multiple sophisticated structural techniques is quite pleasing and generates confidence in the conclusions. The only minor ding is that the final refinement statistics for the high-resolution Fab structure aren't the best. However, the manuscript is worthy of publication as-is.

We thank the reviewer for their appreciation of our study. We have further refined the 1.5Å Ab1303 Fab structure, its R_{free} value is now 0.20. Ab1573 Fab crystal data has minor anisotropy issues, we cut the resolution to 2.5Å and further refined the structure, R_{free} was slightly improved to 0.25. Both structures now have R_{free} values within the acceptable range for the resolution.

Reviewer #3 (Remarks to the Author):

In this manuscript the authors structurally characterize two CD4-binding site (CD4bs) antibodies, isolated from sequentially immunized macaques. Both antibodies exhibited weak, heterologous neutralization. The antibodies recognized a partially open or “occluded” conformation, characterized by cryo-EM. By using DEER spectroscopy, the authors show evidence that an equilibrium between occluded-open and closed Env conformations, suggesting that Ab1303/Ab1573 binding stabilizes an existing conformation.

This study provides useful information on the immune response to commonly used HIV-1 vaccine immunogens, specifically on CD4bs Abs that target partially open Env conformations. The study also provides insights into the conformational heterogeneity of Env immunogens under physiological conditions. The combined use cryo-EM and DEER spectroscopy shows that the antibodies AB1302 and AB1573 bind a conformation of Env that exists is sampled within unliganded populations of these Env immunogens. That these antibodies neutralize HIV-1, albeit weakly, suggests that this Env conformation is also sampled on native virus. The paper is well-written and easy to read. The neutralization data for these antibodies has been submitted for review in a separate manuscript (reference #19).

We thank the reviewer for their appreciation of our study. Also, reference #19 is now published in Science Translational Medicine (<https://www.science.org/doi/10.1126/scitranslmed.abk1533>).

Comments for authors:

1. Figure 1C shows SEC-MALS profiles of the antibody complexes at 25C and 37C, which forms the basis for the conclusion of altered stoichiometry. Given the importance of these conclusions in the context of this study, an orthogonal measure of stoichiometry, such as from negative stain electron microscopy (NSEM), a technique that should be easily accessible to this group, will make the results stronger. Aside from visualizing the stoichiometry of antibodies bound to Env, it will also be interesting to see if any differences are due to altered Env conformations at the different temperatures – these should be revealed even at the lower NSEM resolutions.

In our original experiments, we incubated the Fab-Env samples at 22°C. For example, for Ab1303-BG505, 2D classes of the sample prepared after a 22°C incubation showed that there was one Fab bound per trimer (see figure for reviewers below: Fab densities highlighted in red circles, note that the hole between the Fab V_HV_L and C_HC_L regions are visible from some orientations). The one Fab per Env trimer stoichiometry observed in this experiment was consistent with the stoichiometry we subsequently observed by SEC-MALS. Unfortunately, we were not able to obtain high-resolution 3D reconstructions from any samples prepared after a 22°C incubation. We therefore used SEC-MALS to explore ways to prepare samples for obtaining higher-resolution structures, including incubating samples at 37°C. By SEC-MALS, we showed that the 37°C incubation generated complexes with a Fab:Env stoichiometry approaching 3:1 (Figure 1C in the paper). Cryo-EM results from the 37°C incubation showed that the particles were more homogeneous, and the data eventually produced the structures reported in the paper.

1x Fab bound

2. *There are no methods associated with the SEC-MALS experiments and it is unclear what ratio of Env:antibody was used for these experiments.*

We apologize for this mistake. The methods for SEC-MALS have been added to the revised paper.

3. *The authors show temperature dependent differences in the antibody-bound complexes and show greater Env mobility at 37C. The cryo-EM samples were prepared by incubating at 37C for 2 hrs, although later the sample was exposed to lower temperature ~12C in the Vitrobot for grid plunging. Could the complex change between 37C and 12C? This seems like a relevant question for this study where temperature-dependent Env differences are observed. This can be determined relatively easily by incubating the antibody-Env complex first at 37, and then at 12C followed by assessing stoichiometry by SEC-MALS and NSEM.*

We found that the 12°C temperature helped reduce ice on the grids and prevented fog building up in the chamber. For sample freezing, multiple pairs of tweezers were alternatively used, which were each heated to 37°C before picking up a new grid. The time between loading a sample drop onto a grid and plunge freezing was only a few seconds. Although it is possible that the sample temperature might drop slightly due to the cooler air in the chamber, it seems unlikely that this procedure would alter the complex conformation, especially since the tweezers had been preheated to 37°C. In addition, the portion of the sample on the drop surface that would be in contact with cooler air would be removed during blotting. Given that this procedure could not be reproduced in an SEC-MALS experiment, we do not think doing an SEC-MALS run at 12°C would be informative. We also do not believe that NSEM with a 12°C step would be informative because we would not be using a Vitrobot. In conclusion, although we acknowledge a long exposure to 12°C could result in alternative conformations, we think it is highly unlikely that the procedure we used in the Vitrobot that may have exposed samples to a potentially lower temperature for a few seconds would alter the 3 Fabs/1 Env trimer stoichiometry and conformation observed by EM, which is consistent with the SEC-MALS experiment also conducted at 37°C.

4. *At the end of the first Results para, the authors state: “Here we investigated Ab1303 and Ab1573, which unexpectedly recognized the CD4bs rather than the V3-glycan patch that was*

targeted in the sequential immunization scheme.” It is unclear from the information provided why this result is “unexpected”. Was the CD4 binding site in the immunogens blocked in some way to prevent antibody access? If not, it is not really unexpected that the immune system should elicit antibodies targeting the CD4 binding site in the immunogen.

The CD4bs of the Env trimers we used as immunogens were not altered for germline targeting. Since “wt” BG505 from which the RC1-4fill immunogen was derived and other “wt” SOSIPs do not elicit neutralizing antibodies targeting the CD4bs, we thought it was surprising that we elicited CD4bs-directed neutralizing antibodies using our V3-targeting immunization protocol. As described in the *Science Translational Medicine* paper, the predominant epitope that was targeted in rabbits and NHPs using our prime-boost strategy was the V3-glycan patch, not the CD4bs.

5. It is interesting that the two antibodies described in this study are not as mutated as broadly neutralizing CD4bs antibodies (such as VRC01 and others). Could this be due to reduced structural barriers for antibody access in the partially open Env conformation recognized by these antibodies? Perhaps the authors could comment on this in the discussion.

As now noted in the Introduction, heavily mutated bNAbs arise after several years of HIV-1 infection in humans, including viruses that evolve by mutation. By contrast, our prime-boost protocol does not involve mutating viruses and takes place over a shorter time period. We believe these differences are more likely to account for the lower degree of SHM observed for the CD4bs antibodies described in this paper than the occluded-open conformation that they recognize, noting that the V3-targeting antibodies we reported in the *Science Translational Medicine* paper were also less mutated than bNAbs derived after HIV-1 or SHIV infection.

6. In the methods section “florinated octylmaltoside” should be “fluorinated octylmaltoside”.

Thanks for catching this mistake, which has now been corrected.

7. It is not clear from the figures and text how this antibody interacts with the CD4 binding loop. Would be good to add a figure showing interaction with this key Env region.

Thank you for this suggestion. We’ve updated Fig. 3b and 3d to include this information.

8. The authors mention: “While the Ab1303 light chain was adjacent to the Asn276gp120 glycan, the Ab1573 heavy and light chains contacted the N-glycans attached to Asn197gp120, Asn276gp120, Asn363gp120, and Asn386gp120, likely resulting in glycan rearrangements to allow binding” It will be great to include a figure that shows the glycan rearrangements, by comparing to an Env structure that does not have an antibody bound in the vicinity of these glycans (such as PDB 5FYL).

Thank you for this suggestion. A new figure, Extended Data Fig. 3, shows this comparison.

We thank the reviewers for their careful reading of our paper and hope that the revised version is now suitable for publication in *Nature Communications*.

Sincerely,
Pamela J. Bjorkman

Reviewer comments, second round review:

Reviewer #1 (Remarks to the Author):

The authors have adequately addressed my previous concerns. I have no additional comments.

Reviewer #3 (Remarks to the Author):

The authors have adequately responded to my (and other) reviewers' critiques.